# Deletion of Aurora kinase A prevents the development of polycystic kidney disease in mice

Ming Shen Tham [1,2,5], Denny L. Cottle [1,2,5,6] ✉, Allara K. Zylberberg[1,2,5], Kieran M. Short [1,2], Lynelle K. Jones[1,2], Perkin Chan[1,2], Sarah E. Conduit [3,4], Jennifer M. Dyson[3,4], Christina A. Mitchell [3,4] & Ian M. Smyth [1,2,4,6] ✉

Aurora Kinase A (AURKA) promotes cell proliferation and is overexpressed in different types of polycystic kidney disease (PKD). To understand AURKA's role in regulating renal cyst development we conditionally deleted the gene in mouse models of Autosomal Dominant PKD (ADPKD) and Joubert Syndrome, caused by Polycystin 1 (*Pkd1*) and Inositol polyphosphate-5-phosphatase E (*Inpp5e*) mutations respectively. We show that while *Aurka* is dispensable for collecting duct development and homeostasis, its deletion prevents cyst formation in both disease models. Cross-comparison of transcriptional changes implicated AKT signaling in cyst prevention and we show that (i) AURKA and AKT physically interact, (ii) AURKA regulates AKT activity in a kinase-independent manner and (iii) inhibition of AKT can reduce disease severity. AKT activation also regulates *Aurka* expression, creating a feed-forward loop driving renal cystogenesis. We find that the AURKA kinase inhibitor Alisertib stabilises the AURKA protein, agonizing its cystogenic functions. These studies identify AURKA as a master regulator of renal cyst development in different types of PKD, functioning in-part via AKT.

Polycystic kidney disease (PKD) is a common feature of ciliopathies; a family of disorders associated with defective primary cilium structure and/or function[1,2]. This disease spectrum ranges from conditions like Autosomal Dominant and Recessive PKD (ADPKD, ARPKD), where renal cyst development is the most prominent pathology, through to complex syndromic conditions. Cysts arise from the epithelium of the nephrons and collecting ducts and progressively ablate surrounding tissue, often leading to end-stage renal disease[3]. Genetic characterisation of ciliopathy patients has identified causative mutations in genes involved in assembling and disassembling the primary cilia, regulating ciliary protein transport and in the regulation of cilia-associated cell signalling.

Increasing evidence suggests that two different pathways are involved in the development of PKD, referred to as the major and minor cystogenic pathways[4]. The major pathway is mediated by the POLYCYSTIN proteins which are mutated in Autosomal Dominant PKD (ADPKD)[4], the most prevalent lethal monogenic disease (frequency of ~1/1000[5]). Polycystins localise to the primary cilium forming a channel complex which responds to fluid shear stress[6]. Their deletion dysregulates multiple signalling pathways important for homoeostasis of renal epithelia, causing enhanced proliferation[7–9]. The minor pathway is associated with the genes mutated in the syndromic ciliopathies such as Joubert Syndrome (JS), a rare disorder characterised by developmental malformation of the central nervous system, face and

[1]Development and Stem Cells Program, Monash Biomedicine Discovery Institute, Monash University, Clayton, VIC, Australia. [2]Department of Anatomy and Developmental Biology, Monash Biomedicine Discovery Institute, Monash University, Clayton, VIC, Australia. [3]Cancer Program, Monash Biomedicine Discovery Institute, Monash University, Clayton, VIC, Australia. [4]Department of Biochemistry and Molecular Biology, Monash Biomedicine Discovery Institute, Monash University, Clayton, VIC, Australia. [5]These authors contributed equally: Ming Shen Tham, Denny L. Cottle, Allara K. Zylberberg. [6]These authors jointly supervised this work: Denny L. Cottle, Ian M. Smyth. ✉e-mail: denny.cottle@monash.edu; ian.smyth@monash.edu

limbs and by variably penetrant PKD[10]. JS is genetically diverse, with causative mutations found in more than twenty genes (reviewed in[11]) including *Inositol polyphosphate-5-phosphatase E* (*INPP5E*)[12–15]. INPP5E is a key regulator of AKT signalling[12] and has been proposed to play a role in regulating the minor cystogenic pathway with its ciliary localisation regulated by ADP ribosylation factor-like GTPase 13B (ARL13B), a further ciliopathy protein[16]. Our knowledge of the components of these pathways and their interactions remains incomplete, although it is known that mutations in minor pathway components reduce disease severity when the major pathway is perturbed[4,17]. The factor(s) which potentially regulate the crosstalk between these pathways are unclear.

One possible candidate for such a mediator is Aurora kinase A (AURKA), a protein first identified as a regulator of mitosis[18] and which controls the microtubule cytoskeleton, mitochondrial dynamics and the stability of the primary cilia[19,20]. AURKA is essential for early embryonic development[21] and its amplification has been associated with a number of cancers for which AURKA kinase inhibitors are being employed in clinical trials[22,23]. In breast cancers, AURKA has been shown to cause resistance to PI3K/AKT/mTOR inhibitors by reactivating AKT through phosphorylation at Serine 473[24]. Activation of AURKA's kinase function is linked to phosphorylation of residue T288[25,26] which promotes cell proliferation by coordinating cilia resorption, DNA-damage checkpoints, S phase entry and mitosis[18,19,27,28]. A role for AURKA in renal cyst development is suggested by its over-expression in models of Joubert Syndrome[15] and Autosomal Dominant PKD (ADPKD)[29]. In cell culture models, INPP5E interacts with AURKA[15] and the protein also regulates calcium signalling by interacting with POLYCYSTIN2 (PC2)[29,30]. These findings raise the possibility that AURKA is associated with both the major and minor cystogenic pathways.

Previous studies have examined the role of AURKA in cyst development using Alisertib (MLN8237), an AURKA kinase inhibitor which blocks ATP binding and prevents T288 phosphorylation[31]. Despite evidence supporting a role for AURKA in promoting cyst development, its inhibition using Alisertib instead exaggerates ADPKD in mouse models[32,33]. To address this apparent paradox, we conditionally deleted *Aurka* in mouse models of JS and ADPKD. Despite previous studies suggesting a critical role for AURKA in the regulation of normal mitotic progression[21], we find that *Aurka* is dispensable for the development of the kidney's collecting duct system and for renal homoeostasis. We show that *Aurka* is an obligate driver of cyst formation in neonatal JS and ADPKD models and that its deletion significantly ameliorates adult-onset ADPKD. Transcriptional profiling and phospho-immuno AKT protein analysis of these models suggested shared AURKA-mediated dysregulation of AKT signalling. We then found that AURKA and AKT interact in diseased tissue in models of both ciliopathies and that the striking prevention of cyst formation observed upon co-deletion of *Aurka*, is associated with normalisation of AKT signalling, which occurs independent of AURKA's kinase activity. Consistent with previous studies of ADPKD models, we find that treatment with the AURKA kinase inhibitor Alisertib increases cyst number in JS mice and that this is linked to an unexpected accumulation of AURKA following drug treatment and AKT activation. Cystic disease in both murine models of PKD was constrained by inhibition of AKT. These studies establish AURKA as a central, shared, component of different cystogenic pathways in PKD and show that the protein is important for driving renal cyst formation, at least in part via enhanced AKT signalling.

## Results

### Aurka knock-out prevents cystogenesis

To examine *Aurka*'s role in regulating renal epithelial homoeostasis we crossed *Aurka* floxed mice (*Aurka^{f/f}*)[21] with *B6.Cg-Tg(Hoxb7-cre)Amc* animals (*cre*)[34] to inactivate the gene in collecting ducts (*Aurka^{Δ/Δ}*). In control kidneys, expression of AURKA was detected infrequently at

postnatal day 4 (P4) (Fig. S1a). Gene deletion in *Aurka^{Δ/Δ}* mice was confirmed by immunostaining and PCR analysis, and the animals were born at normal Mendelian ratios, with no evidence of renal pathology at P0, P21, and P150 (Fig. S1b, c, d). Whilst their kidneys were slightly smaller at P0, this difference was not found later (Fig. S1e) and the only abnormality seen was a low incidence of hydronephrosis (Fig. S1f, 7%, $n = 68$). Such animals were excluded from further study. Analysis of ureteric bud morphogenesis by Optical Projection Tomography[35] at embryonic day 14.5 (E14.5) found no overt structural differences in ureteric bud branching (Fig. S1g) or in other measures of ureteric tree morphology (Fig. S1h–l). Taken together, these results demonstrate that AURKA is largely dispensable for the development and homoeostasis of the renal collecting duct system, despite the high levels of cell proliferation in this tissue[36].

To establish an in vivo model of JS in which to study the role of AURKA, *Inpp5e* floxed mice (*Inpp5e^{f/f}*)[13,14] were crossed with *Hoxb7-cre* animals to generate progeny lacking *Inpp5e* in collecting ducts (*Inpp5e^{Δ/Δ}*). Consistent with previous studies using the renal epithelia-specific *Ksp-cre*[14], we observed rapid and comparatively more severe development of collecting duct-derived cysts (Fig. 1a) which limited animal survival to 17-21 days. By P21 AURKA expression was limited to a rare collecting duct cells in wild-type mice, but cyst development was associated with an increase in cells expressing AURKA (Fig. S1m). Mice were then crossed to generate *Inpp5e^{Δ/Δ}; Aurka^{Δ/Δ}* animals and intermediate genotypes. *Inpp5e^{Δ/Δ};Aurka^{Δ/+}* mice were indistinguishable from *Inpp5e^{Δ/Δ}* mice (Fig. 1b–d). In contrast to *Inpp5e^{Δ/Δ}* mice, double mutant kidneys exhibited normal size and outward appearance (Fig. 1a). At P21, the cystic index of the organ was significantly reduced (Fig. 1b) and cyst number decreased by ~90% (Fig. 1c, from averages of 381 to 38). While a small number of residual cysts of a size equivalent to those of *Inpp5e^{Δ/Δ}* mice were observed in double mutant mice (Fig. 1d), these exhibited persistent expression of AURKA (Fig. S1m, n), indicating that these cysts derive from cells in which *Aurka* has not been deleted. Analysis of the numbers of AURKA+ cells in mice of different genotypes confirmed cre activity and reflect the presence of breakthrough cysts characterised by persistent AURKA expression (Fig. S1n). *Cre* only and other control animals exhibited no cysts.

To investigate the capacity of *Aurka* deletion to prevent the development of PKD over time, we analysed *Inpp5e^{Δ/Δ}; Aurka^{Δ/Δ}* mice up to 150 days of age (Fig. 1e). Staining with markers to different nephron segments found that most cysts had collecting duct identities consistent with cre expression (Fig. S1o). Double mutant animals maintained normal kidney-to-body-weight ratios (Fig. 1f) with no increase in the number of cysts (Fig. 1g), cystic index (Fig. 1h) or size (Fig. S1p). Over time the number of cysts reduced slightly as did the cystic index, possibly as a result of eventual Cre mediated *Aurka* deletion, a question we formally examine later in this article. Assessment of renal function by measurement of blood urea nitrogen (at P11, ~60 and ~150; Fig. 1i) or urinary albumin creatinine ratios at P21 (Fig. S1q) showed a complete normalisation of these parameters relative to controls. These studies propose *Aurka* an obligate regulator of renal cyst development in Joubert Syndrome.

We next examined whether AURKA also contributes to cyst development in ADPKD downstream of POLYCYSTIN mutations. *Pkd1^{Δ/Δ}* mice were generated using the same *Hoxb7-cre* driver, resulting in extensive collecting duct-derived cysts (consistent with other studies[37]) requiring euthanasia at postnatal day 11 (P11)(Fig. 1j). Consistent with prior studies[32], these structures were associated increased expression of AURKA (Fig. S2a–e). *Pkd1^{Δ/Δ}* kidneys had significantly increased cystic index (Fig. 1k), cyst number (Fig. 1l), and size (Fig. 1m). Conditional deletion of *Aurka* in *Pkd1^{Δ/Δ}* mice (confirmed by reduction in AURKA expression (Fig. S2c)) resulted in an almost complete suppression of cyst formation. Double mutant kidneys were of similar size and outward appearance to controls (Fig. 1j) and had a reduced cystic index (Fig. 1k) driven by a 98.9%

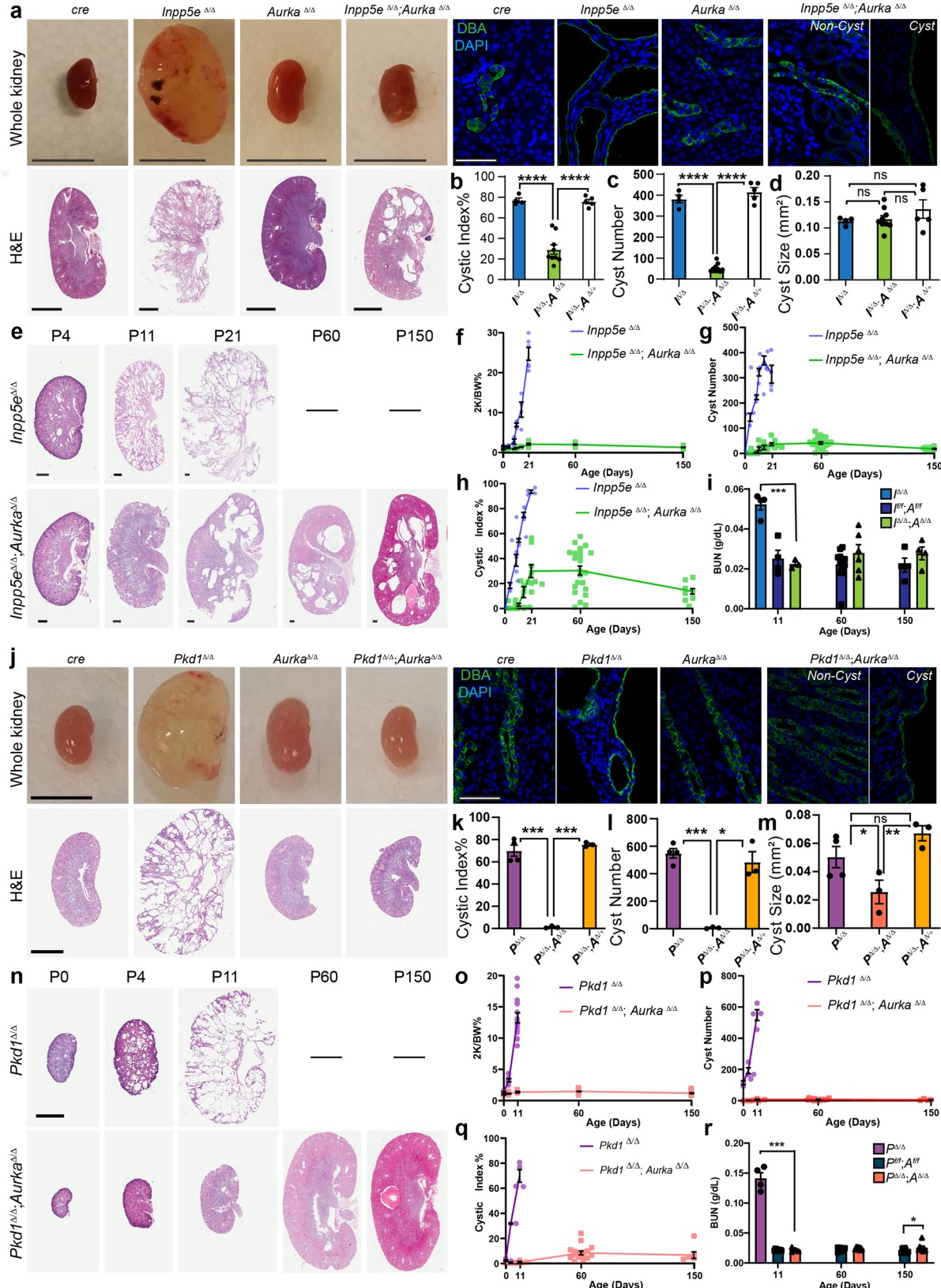

decrease in cyst number (Fig. 1l) and a reduction in the size of the few remaining cysts (Fig. 1m). *Pkd1^{Δ/Δ};Aurka^{Δ/+}* mice were indistinguishable from *Pkd1^{Δ/Δ}* mice (Fig. 1k–m) and the few residual cysts in double mutant kidneys again demonstrated persistent expression of AURKA (Fig. S2d), indicating that they likely arise from incomplete *cre* activity.

To examine the long-term effects of *Aurka* deletion on ADPKD progression, we examined *Pkd1^{Δ/Δ};Aurka^{Δ/Δ}* at P11, P60 and P150 and found them to be of normal size and outward appearance relative to controls at all ages (Fig. 1n). Other parameters such as kidney-to-body-weight ratio, cyst quantity and cystic index did not vary with age (Fig. 1o–q). Remnant cysts expressed collecting duct markers (Fig. S2f)

**Fig. 1 | Aurka knock-out prevents cystogenesis. a, j** Images of mouse kidneys from indicated genotypes (bar = (A) 1 cm (whole organs); **j** = 0.5 cm; 2 mm (for H&E sections); 50 μm (for immunostains). **b, k** cystic index (*InppSe p* values for pairs indicated from left to right; $p = 8.7 \times 10^{-7}$, $p = 9.3 \times 10^{-7}$ and *Pkd1 p* values; $p = 3.9 \times 10^{-4}$, $p = 4.1 \times 10^{-6}$). **c, l** average cyst number per cross-section number of cysts (*InppSe p* values for pairs indicated from left to right; $p = 4.6 \times 10^{-5}$, $p = 7.5 \times 10^{-6}$, and *Pkd1 p* values; $p = 2.6 \times 10^{-4}$, $p = 0.0122$). **d, m** average cyst cross-sectional size (*Pkd1 p* values for pairs indicated from left to right; $p = 0.0419$ and $p = 0.0094$). **e, n** Kidney sections of indicated genotypes of increasing age. **e** bars = 0.4 mm and **n** = 4 mm. **f–h** and **o–q** Plot of kidney-to-body-weight ratio, cyst number and cystic index of indicated mice over time. **i, r** Blood Urea Nitrogen (BUN) assessment of kidney function with age (*InppSe p* values indicated = 0.0003, *Pkd1 p* values for pairs indicated from left to right $p = 0.0005$, $p = 0.0490$). All graph data indicate mean ± S.E.M. **a–e** Mice were P17–P21 days of age. **j–o** Mice were P11 days of age. Graph abbreviations that apply throughout *I InppSe, P Pkd1, A Aurka*, ns not significant. **a–d** $n = 4$–9, **e–h** $n = 3$–24, **i** $n = 4$–12, **j–m** $n = 3$–4, **n, p–q** $n = 3$–16, **o** $n = 3$–34, **r** $n = 4$–9. Cysts were not present in the kidneys of other mouse genotypes and are omitted from cyst size, number and index graphs. Additional data in Figs. S1 and S2. Exact n and data point values provided in supplementary material- Source Data File.

and increased slightly in size over time (Fig. S2g). Measurements of blood urea nitrogen (BUN, Fig. 1r) in double mutant kidneys were normal at P11 and P60 but exhibited a modest increase at P150. Overall, these experiments show that *Aurka* deletion prevents renal cystogenesis caused by loss of *InppSe* or *Pkd1*.

## Aurka knock-out normalises ciliation, proliferation and DNA-damage responses

Alterations in ciliation, polarity and cell proliferation are associated with renal cyst formation[14,15,38,39]. Confocal analysis found no obvious changes in collecting duct adhesion and polarity from KSP immunostaining of *InppSe^{Δ/Δ}* kidneys at P21, *Pkd1^{Δ/Δ}* kidneys at P11 and their rescued counterparts (Fig. S2h). Furthermore, no significant changes were observed in cilia structure in renal epithelial cells in *InppSe^{Δ/Δ}* mice at P4 (Fig. 2a, S2i). Cystic cells also displayed enhanced proliferation (Fig. 2b. S2j), however, no change in cilia length at P21 was observed (Fig. 2c). Collecting duct cell ciliation in control mice decreased from ~100% to ~80% between P0 and P21 (Fig. 2d). Studies in murine *InppSe^{-/-}* fibroblasts indicate that the protein regulates cilia disassembly in response to growth factor stimulation[40], but the level ciliation at birth in *InppSe^{Δ/Δ}* collecting duct epithelia was indistinguishable from controls (Fig. 2d). However, ciliation decreased at the onset of cyst initiation at P4, paralleling the increase in cell proliferation (Fig. 2b, d). Co-deletion of *Aurka* largely normalised cell proliferation and ciliation (Fig. 2d) aside from a small reduction in comparative cilia number at P21 (Fig. 2d). It is likely that this reduction is due to loss of INPP5E and its broader role in maintaining cilia stability independently of AURKA's actions[41]. Similar analysis of *Pkd1^{Δ/Δ}* kidneys found no changes in cilia structure (Fig. 2e), proliferation (Fig. 2f) or length (Fig. 2g) at birth, while the reduction in ciliation and increase in cell proliferation characteristic of cyst progression were both normalised by co-deletion of *Aurka* (Fig. 2h). DNA damage has been reported in ciliopathy models[42] and increased γ-H2AX expression is a feature of *InppSe* and *Pkd1* mutant kidneys[43]. However, co-deletion of *Aurka* normalised this marker in both JS (Fig. 2i, j, S2k) and ADPKD models (Fig. 2k, S2k, S2i). Examination of ciliation and expression of Ki67 and γH2AX in break-through cysts of both *InppSe* and *Pkd1* mice confirmed cellular phenotypes similar to those arising from deletion of each gene alone, further supporting the idea that they derive from cells escaping Cre mediated deletion of *Aurka* (Fig. S2m–r). Overall, these results show that co-deletion of *Aurka* normalises cyst associated differences in cell ciliation, proliferation and DNA damage in both *InppSe^{Δ/Δ}* and *Pkd1^{Δ/Δ}* mice.

## Aurka deletion reduces the severity of adult-onset ADPKD

The rate of cyst formation differs between neonatal and conditional late onset models of ADPKD, a finding attributed to a shift in organ metabolism at postnatal day 13[44]. To examine whether *Aurka* deletion could also ameliorate cyst development after this shift in adult mice (more analogous to ADPKD in humans), we employed a doxycycline-inducible model of ADPKD[17] which employs *Pax8-rtTA* and *TetO-Cre* transgenes to mediate gene knockout in the collecting duct, proximal and distal tubules of the nephron. ADPKD was induced at 4 weeks of age and kidneys were collected at 12, 18 and 20 weeks of age, at which

point advanced cystic kidney disease had developed in mice lacking *Pkd1* (Fig. 3a). Analysis of cyst size, number and cystic index identified significant improvements in all three measures induced by co-deletion of *Pkd1* and *Aurka* (Fig. 3a–d). although improvements in kidney-to-body-weight ratios and Blood Urea Nitrogen (BUN) measurements, which vary considerably with disease progression at these earlier time points, were not apparent (Fig. S3a, b). In contrast to early onset PKD induced by *Hoxb7-cre* mediated *Pkd1* deletion (see Fig. 1k–m), cyst development was also attenuated by deletion of a single copy of *Aurka*. It was noted this phenomenon was less evident in proximal tubules, which may relate to differences in AURKA expression between tubule types (Fig. S3c–e). While mice lacking *Pkd1* rapidly succumbed to disease by the ethically allowed limit of the experiment (20 weeks of age), we found that both hetero- and homozygous deletion of *Aurka* significantly improved the survival in a dose dependent manner and this correlated with partial restoration of kidney function reflected in BUN ratios at experimental end points (defined as the conclusion of the 20-week time course or at the point of humane euthanasia if earlier) (Fig. 3e).

While our initial experiments showed a critical pathogenic role of *Aurka* when temporally co-deleted with either *InppSe* or *Pkd1*, we sought to examine the impact of *Aurka* deletion in established cysts. To do so we utilised the *Pkd1 RC* model which includes a hypomorphic *Pkd1* variant first identified in an ADPKD patient[45]. Initial assessment of disease (via kidney-to-body-weight ratios) in this model indicated existing disease at postnatal day 28, a slow progression of disease up to 6 months of age without significant sex difference, with sex divergence from 6 to 9 months of age (Fig. S4a). After allowing significant cyst development in this model, we then deleted *AURKA* at 4 weeks (P28) of age using the Dox system previously employed to study late onset disease, ageing mice to 26 weeks (6 months) of age. This maximum time point therefore facilitates pooling of sexes as it proceeds the sex divergence. Assessment of cystic phenotypes at this point demonstrated that *Aurka* deletion had completely halted cyst growth (Fig. S4b–f). There was no overt decrease in cyst number in this experiment, as it is likely the early events precipitating cystogenesis have already occurred at the point at which *Aurka* was deleted. Furthermore, the modest gain in cyst quality over time more likely reflects cysts that were present at P28 but below the detection threshold.

## AURKA alters the AKT signalling pathway

To profile the molecular drivers of cyst development curtailed by *Aurka* deletion, we performed RNA sequencing of ADPKD and JS cystic kidneys at P4 (to preclude changes associated with advanced disease) with and without concomitant *Aurka* deletion. 485 significant gene expression changes were found in *InppSe^{Δ/Δ}* kidneys relative to *cre* controls but only 40 were apparent upon co-deletion of *Aurka* (Fig. 4a). This magnitude of this normalisation suggests that INPP5E and AURKA act closely at a functional level. Although the expression of 120 genes was changed in *Aurka^{Δ/Δ}* kidneys relative to cre only controls, these did not enrich any KEGG signalling pathways or obvious PKD-related genes. Analysis of *Pkd1^{Δ/Δ}* samples identified 1532 dysregulated genes relative to *cre* controls (Fig. 4b) but unlike our JS model, this number was only modestly corrected by *Aurka* co-deletion (to 1252 genes)

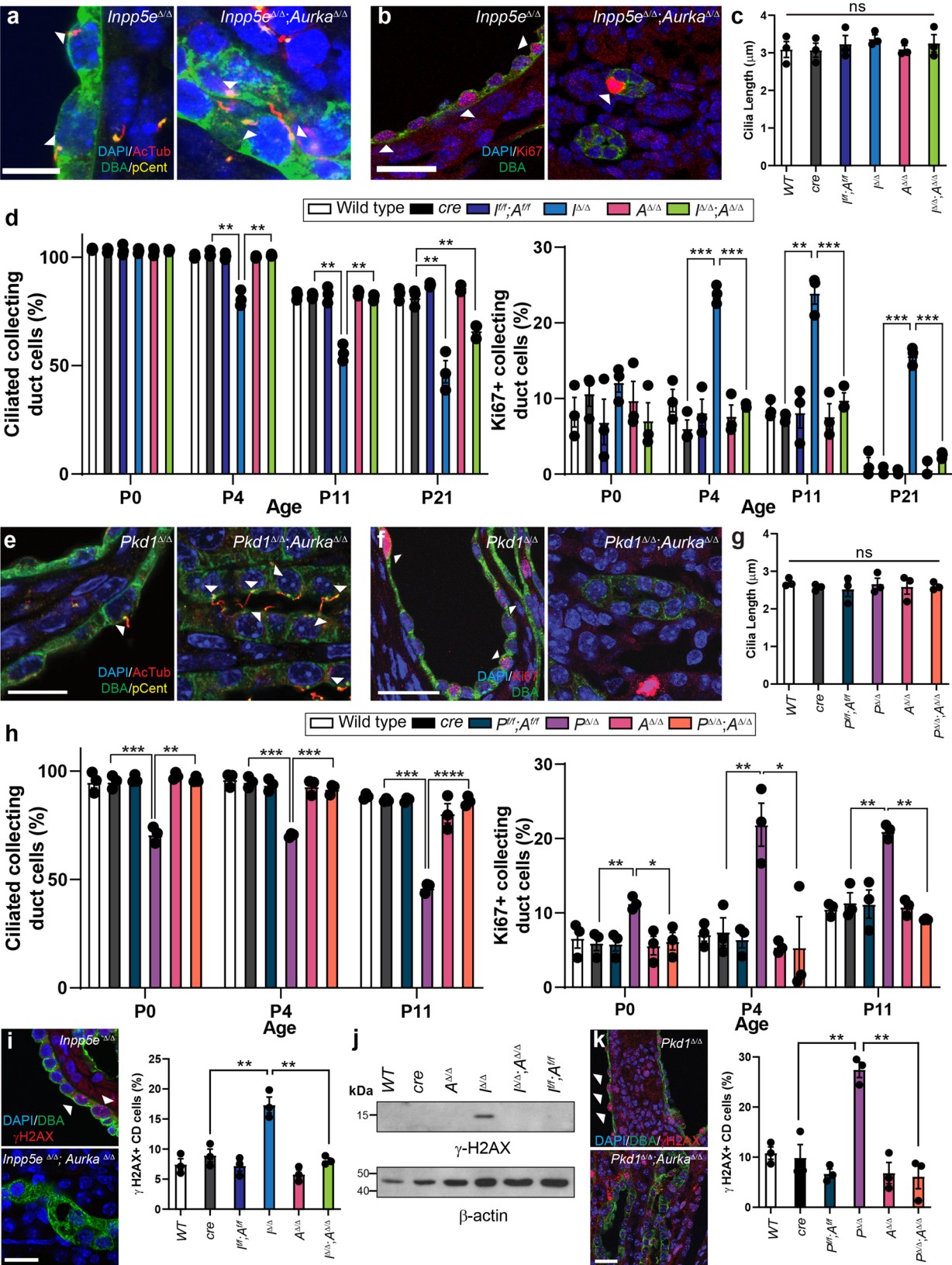

(Fig. 4b), despite the rescue in cystic disease phenotypes. A full list of differentially expressed genes is included in Supplementary Data 1. To determine whether common *Aurka*-dependent drivers of cystogenesis were apparent in our JS and ADPKD models, we compared changes in KEGG pathways in single mutants (*Inpp5e* or *Pkd1*, versus *cre* controls) and double mutants (*Inpp5e; Aurka* or *Pkd1; Aurka* versus single

mutants) (Fig. 4c, Supplementary Data 2). We identified five shared dysregulated pathways: Pathways in cancer, MAPK, Breast cancer, Gastric cancer and, notably, the AKT signalling pathway. These results suggest that while the individual genes modulated by AURKA differ in both PKD models, they feed into a small subset of common signalling pathways.

**Fig. 2 | Aurka knock-out normalises ciliation, proliferation and DNA-damage responses. a, e** P4 kidney sections immunostained for DAPI, acetylated-tubulin (AcTub), Pericentrin (Pcent) and DBA. Arrowheads indicate collecting duct cells with cilia. Bar = 12 μm (*InppSe*), 10 μm (*Pkd1*). **b, f** P4 kidney sections immunostained for DAPI, Ki67 and DBA. Arrowheads demonstrate collecting duct cells positive for Ki67. Bar = 25 μm. **c, g** Quantification of collecting duct cilia length at P21 in *InppSe* model and P11 in *Pkd1* model, respectively (ns not significant). **d, h**-left Quantification of the proportion of collecting duct cells exhibiting cilia across the time points indicated. *InppSe p* values for pairs indicated from left to right; $p = 0.0021$, $p = 0.0047$, $p = 0.0020$, $p = 0.0016$, $p = 0.0056$ and $p = 0.0039$. *Pkd1 p* values for pairs indicated from left to right; $p = 0.0006$, $p = 0.0017$, $p = 0.0008$, $p = 0.0008$, $p = 0.0002$, $p = 0.00001$. **d, h**-right Quantification of the proportion of collecting duct cells stained for Ki67 across time. *InppSe p* values for pairs indicated from left to right; $p = 0.0002$, $p = 0.0007$, $p = 0.0031$, $p = 0.0013$, $p = 0.0002$ and $p = 0.0005$. *Pkd1 p* values for pairs indicated from left to right; $p = 0.0092$, $p = 0.0171$, $p = 0.0096$, $p = 0.0180$, $p = 0.0045$, $p = 0.0010$. **i, k** P4 Kidney sections immunostained for DAPI, γ-H2AX and DBA; arrowhead indicate CD cells expressing γ-H2AX reactivity. Quantification of P4 γ-H2AX expression in collecting ducts (*InppSe p* values for pairs indicated from left to right; $p = 0.0043$, $p = 0.0069$, *Pkd1 p* values; $p = 0.0042$, $p = 0.0016$). **j** Western blot for relative γ-H2AX expression at P21, re-probed with Actin, Bar = 25 μm. All graph data indicates mean ± S.E.M. $n = 3$ all subpanels. Additional data in Figs. S1 and S2. Full vertical lane scans of WB image (**j**) in Source Data File. Exact data point values provided in supplementary material-Source Data File.

INPP5E regulates AKT activation via degradation of the phosphoinositide PI(3,4,5)P$_3$ which localises AKT to the inner wall of the plasma membrane, allowing its activation by phosphorylation at T308 by PDK1[46] and S473 by mTOR[47,48]. We have previously reported increases in phosphorylation at both residues in cystic epithelium in advanced disease caused by loss of *InppSe*[14] and so sought to determine whether similar changes were evident early in cyst progression. We found that at P4 the number of pAKT (T308)$^+$ collecting duct cells exhibiting was significantly elevated in both *InppSe*$^{\Delta/\Delta}$ and *Pkd1*$^{\Delta/\Delta}$ mice but reduced upon co-deletion of *Aurka* (Fig. 4d, e). *Pkd1*$^{\Delta/\Delta}$;*Aurka*$^{\Delta/+}$ mice exhibited an intermediate reduction in the number of pAKT T308 collecting duct cells (Fig. 4e). Furthermore, partitioning this data by cystic and non-cystic tubules demonstrated the elevation of pAKT (T308) was localised within cystic tubules (Fig. S5a, b). At this early timepoint no changes in pAKT (S473) were noted, although baseline phosphorylation levels were higher than for pAKT (T308) (Fig. S5c). Upregulation of AKT expression, an increase in the AKT pT308/total AKT ratio and downstream pathway activity (via p4EBP1 T37/46) was further confirmed by AlphaLISA assays in the JS murine model and these parameters were normalised upon *Aurka* deletion (Fig. S5d–f). In our ADPKD model, an increase in the AKT pT308/total AKT ratio and downward trend upon *Aurka* deletion was seen by phospho-immunoblotting (Fig. S5g–i). These findings indicate that heightened AKT activity mediated by AURKA is commonly associated with early cyst development driven by *InppSe* or *Pkd1* deletion in collecting duct cells.

Given that *Aurka* deletion apparently modulates AKT activity and correlates with ameliorated cyst development, we examined whether the two proteins might form a physical complex. As commercial antibodies raised to human AURKA did not exhibit sufficient affinity to immunoprecipitate its mouse orthologue, we engineered a c-terminal V5-tag into the endogenous mouse *Aurka* allele using CRISPR technology. The resulting *Aurka*$^{V5/V5}$ mice were viable and healthy, confirming the V5-tag did not impair AURKA function. *Aurka*$^{V5}$ mice were inter-crossed with our neonatal PKD models to generate mice expressing AURKA-V5 from at least one *AURKA* allele. Immunostaining of kidneys showed co-localisation of V5 and AURKA; particularly in cystic *Pkd1*$^{\Delta/\Delta}$;*Aurka*$^{V5/V5}$ and *InppSe*$^{\Delta/\Delta}$;*Aurka*$^{V5/+}$ mice but not in unmodified disease controls (Fig. 4f, g). The *Aurka-V5* allele did not alter the development of PKD in either model (Fig. S5j, k). V5 pulldown experiments enriched multiple AURKA-V5 bands consistent with known N-terminally processed isoforms[49] (Fig. 4h, i) and full-length isoforms were further detected in control testis extracts[50] (Fig. S5l, m). Specific interactions between AURKA-V5 and AKT were observed in cystic *Pkd1*$^{\Delta/\Delta}$; *Aurka*$^{V5/V5}$ and *InppSe*$^{\Delta/\Delta}$;*Aurka*$^{V5/+}$ samples but not in control *Aurka*$^{V5/V5}$, *Aurka*$^{V5/+}$, *Pkd1*$^{\Delta/\Delta}$ or *InppSe*$^{\Delta/\Delta}$ mice (Fig. 4h, i). Addition of competing/blocking V5-peptide countered AURKA-V5 enrichment and blocked AKT co-immunoprecipitation confirming the interaction was mediated by AURKA and not the V5-tag (Fig. 4h, i). To investigate whether AKT interacted solely with N-terminally truncated isoforms of AURKA or could also interact with full-length AURKA, we repeated the AURKA-V5 pulldown experiments with testis

extracts which express only full-length AURKA. Once more, AKT was observed to co-purify with AURKA-V5 (Fig. 4j), indicating that AKT interacts with multiple isoforms of AURKA.

## AURKA regulates pAKT (T308) status and co-localises with PDK1

To further investigate how AURKA might contribute to AKT pathway regulation, sub-confluent mouse collecting duct (mIMCD3) cells were transfected with siRNA to reduce AURKA levels (Fig. 5a, b). This led to relative decreases in pAKT (T308) (but not pAKT (S473)) (Fig. 5c, d) and to modest reductions in total AKT (Fig. 5e). Treatment with the AURKA kinase inhibitor Alisertib resulted in reduced pAURKA (T288)/total AURKA ratio following 48 h of exposure (Fig. S6a) but unexpectedly led to in a 1.5-fold increase in AURKA protein levels (Fig. 5f, g). Elevated AURKA was associated with an increase in pAKT (T308) (Fig. 5h, i) and a modest reduction in total AKT (Fig. 5j). This phenomenon was not observed following 24 h of Alisertib exposure (Fig. S6b), suggesting that activation of pAKT (T308) occurs after longer Alisertib exposure periods, possibly because of AURKA accumulation. To confirm whether over-expression of AURKA could stimulate AKT activity in a kinase-independent fashion, HA-tagged wild type and kinase dead (KD; K162R) AURKA were expressed (Fig. 5k, l). Neither construct demonstrated auto-phosphorylation of T288 (Fig. S6c), however both constructs increased pAKT (T308) (but not S473, Fig. 5m, n) levels with KD increasing total AKT (Fig. 5o). This indicates the activation of AKT involves kinase-independent actions of AURKA. These effects were not observed upon serum starvation (Fig. S6d).

We next employed immunofluorescence to determine where this interaction may take place and found that AURKA and pAKT (T308) co-localised to the cilia base (Fig. 5p), along with PDK1, the principal mediator of AKT(T308) phosphorylation (Fig. 5q). While AURKA was often found alone at this subcellular location, the same was not true for either pAKT or PDK1 which largely co-localised with AURKA at this location (Fig. 5p, q). This dynamic suggests AURKA scaffolds recruitment of a complex of these factors. In contrast, pAKT S473 was largely absent from the basal body (Fig. 5r) and rarely co-localised with AURKA.

Given, the observation that Alisertib acts as an agonist of AURKA's kinase-independent functions, we next tested if other AURKA kinase inhibitors triggered elevated AURKA levels and concomitant increases in AKT pT308. Analogous experiments using MK5108, VX680 and C1368 found that both phenotypes were apparent after 48 h of exposure (Fig. S7a–c).

## Alisertib increases cystogenesis in vivo in JS model

Previous studies had suggested that treatment of an adult-onset ADPKD model with Alisertib resulted in exacerbation of cystic phenotypes[32,33], an observation consistent with the increase in AURKA stability and AKT activation noted in vitro (Fig. 5f–i). To examine whether similar effects were evident in our model of Joubert Syndrome and reflective of a shared cyst promoting mechanism, we treated *InppSe*$^{\Delta/\Delta}$ mice and control animals with the drug from P9 to P13,

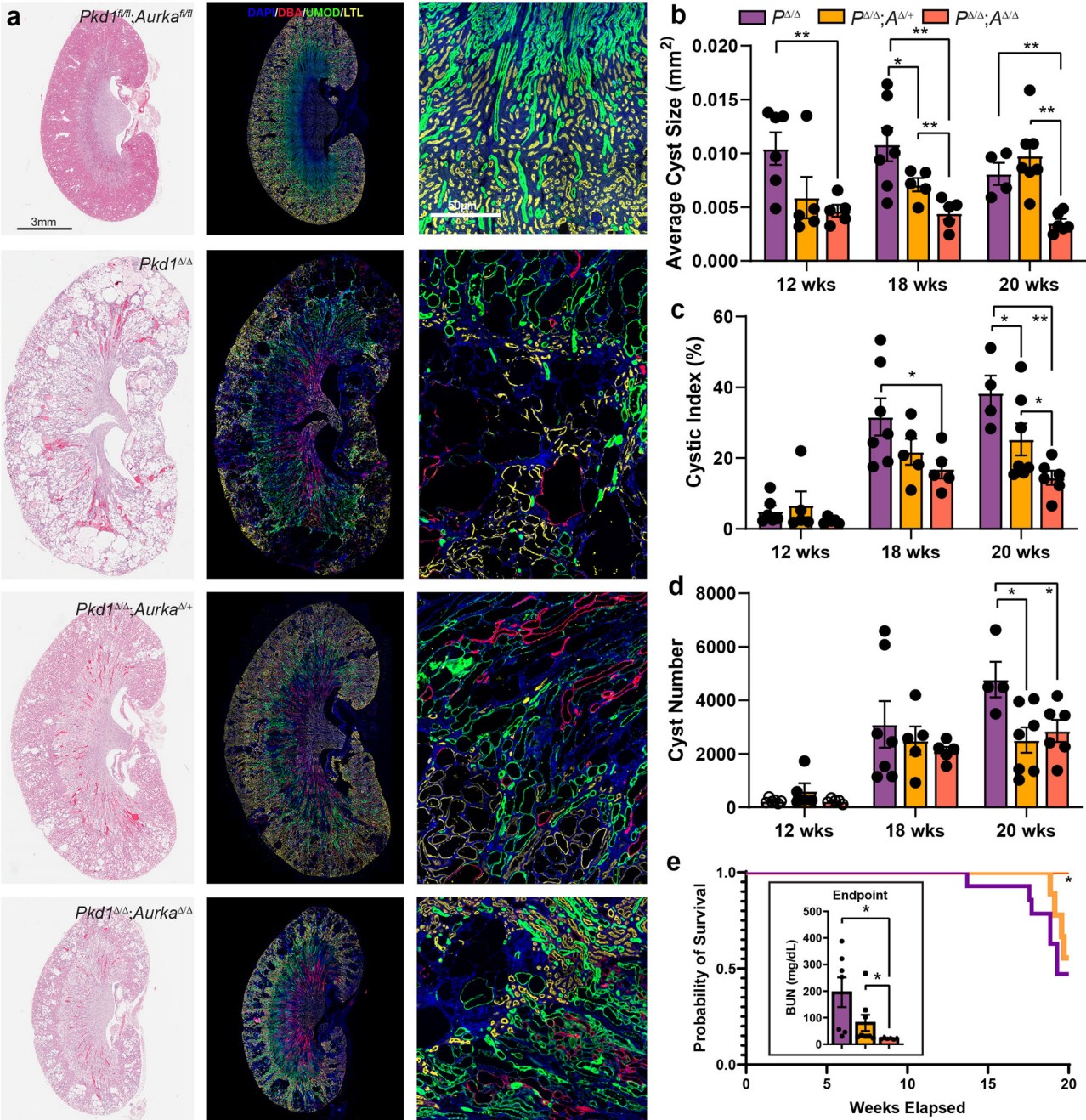

**Fig. 3 | Aurka deletion reduces the severity of adult-onset ADPKD.**
**a** Representative images of whole mouse kidney sections stained for either Haematoxylin and Eosin (H&E, left), or DAPI, DBA, Uromodulin (UMOD) and LTL (middle and right). Kidneys were collected at 20 weeks of age from mice of the indicated genotypes. Scale bar = 3 mm (left and middle), 50 µm (right). The staining intensity of DBA was low in *Pkd1 f/f; Aurka f/f* non-cystic animals compared to cystic animals. **b** The average cyst size per cross-section for overall cyst analysis, significant *p* values indicated from left to right: 0.0054; 0.0282, 0.0027, 0.0084, 0.0077, 0.0001. **c** Analysis of cystic index measured overall; *p* values indicated from left to right: 0.0170, 0.0445, 0.0055, 0.0295. **d** Quantitation of the average cyst number per cross-section overall; *p* values indicated from left to right: 0.0158, 0.0272. **e** Kaplan–Meier curve indicating survival of *Pkd1^{Δ/Δ}*,

*Pkd1^{Δ/Δ};Aurka^{Δ/+}* and *Pkd1^{Δ/Δ};Aurka^{Δ/Δ}* over time. Comparing all three genotypes gives *p* = 0.0491 (Mantel-Cox test, Chi-square 6.029, df 2). Pairwise comparison *p* value between *Pkd1^{Δ/Δ}* and *Pkd1^{Δ/Δ};Aurka^{Δ/Δ}* = 0.0166. Assessment of Blood Urea Nitrogen (BUN) as a measure of kidney function mice of indicated genotypes at experimental end point (either that the conclusion of 20-week time course or at the point of humane euthanasia if earlier, left to right *p* = 0.0108 and *p* = 0.0493). All graphs indicate mean ± S.E.M. (**a**–**d**) *n* = 4–8 at all time points for all genotypes, see exact *n* and data point values provided in supplementary material- Source Data File. Kaplan–Meier curve data displays *n* = 20 (*Pkd1^{Δ/Δ}*), *n* = 19 (*Pkd1^{Δ/Δ}; Aurka^{Δ/+}*) and *n* = 16 (*Pkd1^{Δ/Δ};Aurka^{Δ/Δ}*). Cysts were not present in the kidneys mouse genotypes not indicated. Additional data in Figs. S3 & S4.

collecting tissue at P15 (a period during which cyst number is actively increasing). Alisertib activity was reflected in the hair loss observed in all treated mouse cohorts (but not vehicle, Fig. 6a), a side effect also observed in human trials[51]. While initial analysis of Alisertib-treated *Inpp5e^{Δ/Δ}* mice identified a modest reduction in kidney-to-body-weight ratio and cyst size suggestive of therapeutic benefit (Fig. 6a–c), cystic index was unchanged (Fig. 6d) and cyst number was significantly increased (Fig. 6e). In contrast, *Inpp5e^{Δ/Δ};Aurka^{Δ/Δ}* Alisertib-treated mice displayed no change in kidney-to-body-weight ratio, cyst index, cyst number or cyst size (Fig. 6b–e), confirming Alisertib requires AURKA

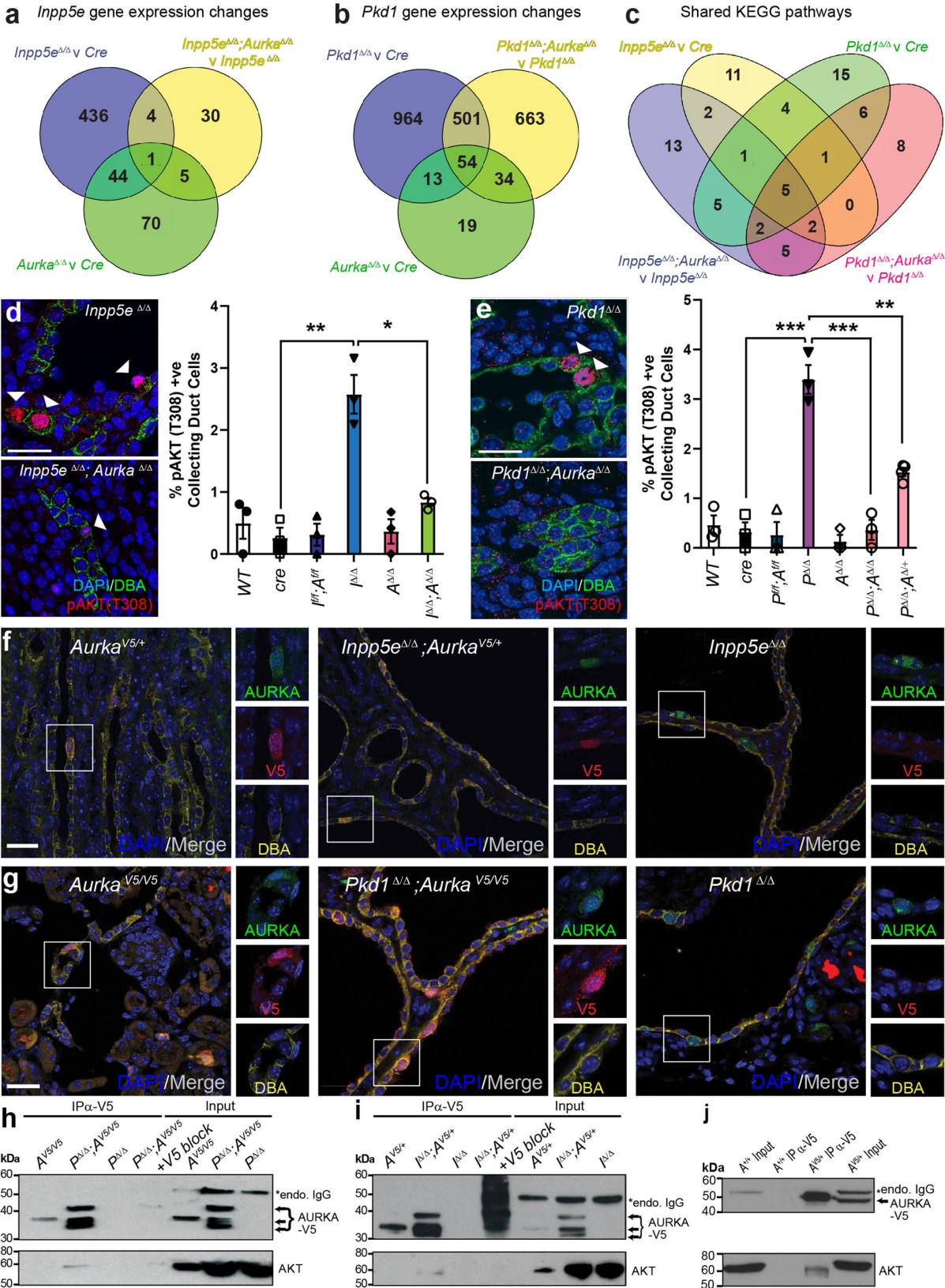

(and not AURKB or other targets) to elicit these pro-cystogenic effects. In vehicle-treated *Inpp5e*[Δ/Δ] kidneys, we observed focal AURKA and AKT co-localisation in cilia and resorbing cilia structures at low frequency, as expected for asynchronous cycling cells (Fig. 6f). However, Alisertib treatment of the same mice markedly reduced ciliated cells (Fig. 6g) but not in *Inpp5e*[Δ/Δ]*;Aurka*[Δ/Δ] animals (Fig. 6g). Alisertib's biological

actions were further confirmed by examining p53, which is specifically inhibited by AURKA's kinase activity[27] and whose nuclear localisation and active form were increased as expected (Fig. 6h, i).

Consistent with our in vitro findings, Alisertib treatment of *Inpp5e*[Δ/Δ] mice increased AURKA (Fig. S8a) and pAKT (T308) positive epithelial cells (Fig. S8b). Double AURKA and pAKT (T308) positive

**Fig. 4 | AURKA alters the AKT signalling pathway.** Venn diagrams show the number of genes whose expression varies by ±1.5-fold in P4 *Inpp5e* (**a**) and *Pkd1* (**b**) mouse models comparing the genotypes indicated. **c** Venn diagram comparing altered KEGG Pathways in P4 kidneys of indicated genotypes. **d**, **e** Staining of P4 kidney sections for DAPI, pAKT (T308) and DBA (arrowheads indicate pAKT T308 expression; bar = 25 μm) and their quantifications. *Inpp5e p* values from left to right; *p* = 0.0035, *p* = 0.0130 and *Pkd1 p* values; *p* = 0.0008, *p* = 0.0008, *p* = 0.0085. **f**, **g** Expression and co-localisation of AURKA-V5 and AURKA in control and cystic kidneys of both models at P15 and P11 respectively (bar = 25 μm). **h** Co-immunoprecipitation of endogenous AKT with AURKA-V5 from P11 *Pkd1*$^{\Delta/\Delta}$ mouse kidneys and competition with V5-peptide (blocking IP) (* = mouse endogenous IgG). **i** Co-immunoprecipitaton of endogenous AKT as in (**h**) from *Inpp5e*$^{\Delta/\Delta}$ kidneys at P15. Additional data in Fig. S5. **j** Co-immunoprecipitation of endogenous AKT from testis extracts. All graphs display mean ± S.E.M. **a**–**c** *n* = 6, **d**, **e** *n* = 3–4, **f**–**j** *n* = 3. Exact *n* and data point values provided in supplementary material- Source Data File.

epithelial cells (Fig. 6j, k) were similar in quanta to the % of pAKT (T308) positive collecting duct cells. AURKA also co-labelled with Ki67 (Fig. 6l and Fig. S8c). These observations confirm that Alisertib-driven stabilisation of AURKA and pAKT (T308) phosphorylation occur together in cyst cells and that these marks correlate with the features of increased cyst burden, increased cilia resorption and hyper-proliferation. Taken together with our findings in mIMCD3 cells, these experiments provide evidence that while Alisertib inhibits AURKA's kinase activity, it is a specific agonist of AURKA's kinase-independent cystogenic function(s).

## Inhibition of AKT constrains cyst formation

Our observations suggest that activation of AKT is an important step in driving renal cystogenesis. To test this, we performed an oral dosing experiment with MK2206, an allosteric AKT inhibitor which impairs phosphorylation of AKT at both T308 and S473[52]. We treated *Inpp5e*$^{\Delta/\Delta}$ pups daily from P9 and harvested on P15 (similar to the time course of previous Alisertib treatment) and found that the kidneys of *Inpp5e*$^{\Delta/\Delta}$ mice treated with 75 mg/kg of MK2206 were significantly smaller than vehicle controls (Fig. 7a, b and Fig. S9a). Moreover, AKT inhibition reduced the cystic index (Fig. 7c), number of cysts (Fig. 7d) and cyst size (Fig. 7e). Treated animals also exhibited increased ciliation (Fig. 7f), a reduction in the proportion of collecting duct cells expressing pAKT (T308) (Fig. 7g, h, S9b) and reduced pAKT (S473) (Fig. S8c), confirming the drugs bioactivity. Notably, the proportion of AURKA+ cells was also reduced by MK2206 (Fig. S9d), an observation consistent with AKT's previously reported transcriptional regulation of *Aurka*[15]. The number of cells co-expressing AURKA and Ki67 was also reduced (Fig. S9e, f).

We repeated this experiment using our early onset ADPKD model, in which disease progression is more rapid. Animals were treated daily with 10, 37.5 or 75 mg/kg of MK2206 from P4 to P10 before sacrifice at P11. These dosages were generally well tolerated, although *Pkd1*$^{\Delta/\Delta}$ mice (but not control animals) treated with 75 mg/kg were runted by the end of the treatment regime. Treated animals displayed a considerable improvement in renal morphology, with more normal renal parenchyma apparent compared to vehicle-treated controls (Fig. 7i). These changes were associated with significant, dosage-dependent, reductions in kidney-to-body-weight ratio (Fig. 7j), cystic index (Fig. 7k), cyst number (Fig. 7l) and cyst size (Fig. 7m). *Pkd1*$^{\Delta/+}$ mice treated with MK2206 did not show any significant changes in renal morphology (Fig. S9g, h). The reduction in disease severity in treated ADPKD animals was associated with corrections in the levels of cell ciliation (Fig. 7n) and a reduction in the number of collecting duct cells co-expressing AURKA, pAKT (T308) (Fig. 7o, p) and Ki67 (Fig. 7q, r). These results confirm the actions of MK2206 on AKT phosphorylation and identify a role for AKT in cyst formation in mouse models of JS and ADPKD.

## Discussion

Aberrant hyper-activation of AURKA correlates with the formation of renal cysts in ADPKD[29] and JS[15]. Paradoxically, studies in mouse models of ADPKD have found that inhibition of AURKA's kinase activity worsens ADPKD[32,33]. Here, we show that *Aurka* deletion significantly and stably prevents cyst formation and/or reduces cyst size caused by loss of *Pkd1*. Moreover, we show that *Aurka* deletion also prevents the development of PKD following loss of *Inpp5e*. While we see slight differences in the magnitude of rescue between these models, closer examination suggests this is a technical limitation due to differing Cre efficiency across the *Pkd1, Inpp5e* and *Aurka* floxed loci. Collectively, these findings indicate that AURKA is a master regulator of both the polycystin-mediated (major) and syndromic (minor; JS) pathways that drive cyst development in different types of PKD. Comparative analysis of these models and subsequent in vivo inhibition studies demonstrate that these effects are mediated, at least in part, by signalling through AKT which we show binds to AURKA in cystic kidneys (Fig. 7s).

The involvement of AURKA and AKT in both types of PKD is intriguing. The regulation of AKT signalling by INPP5E is well described[14] and recent studies have demonstrated direct and functionally relevant interactions between AURKA and INPP5E[15]. The close functional links between these three proteins is also consistent with the normalisation of gene expression changes in the kidney upon co-deletion of *Inpp5e* and *Aurka*. However, the same is not true of *Pkd1* deletion which causes dramatic changes in transcription which are, in large part, unaffected by co-deletion of *Aurka*. Taken in the context of the almost complete prevention of cyst development in both PKD models when *Aurka* is ablated, this finding suggests that despite the differences in transcription, AURKA and AKT are significant mediators of cyst development in both types of disease and potentially a convergence point of different cystogenic pathways.

In animal models, the cystic disease resulting from *Pkd1* gene disruption is aggressive before P13 but relatively mild when gene deletion occurs after this point[44]. Here, we show that AURKA is important in regulating PKD severity in both early- and adult-onset ADPKD models. Interestingly, haploinsufficiency for *Aurka* curtailed disease development in severe adult-onset but not neonatal models of ADPKD nor the mild RC hypomorphic model. AURKA is expressed at very low levels in healthy adult kidneys and higher levels in neonatal organs (Fig. S9i). Because AURKA is required for the formation and/or growth of cysts, the lower levels of gene expression in adult organs may explain the increased sensitivity to gene dosage in the *Pax8-rtTA TetO-Cre* adult model relative to early onset models. In RC mice we speculate the mild nature of the disease along with different cellular dependencies on AURKA with respect to cyst initiation and progression may explain why haploinsufficiency did not afford benefit. Regardless, the prevention of severe adult disease with only a partial depletion of *Aurka* may be important when considering the gene as a target for therapeutic intervention.

With regards to the mechanism by which AURKA might regulate AKT activity during cyst development, we provide evidence that AURKA interacts with AKT in the testis and cystic (but not healthy) kidneys, and that AURKA regulates AKT phosphorylation, independent of its kinase activity (Fig. 7s). A clinically important role for AURKA in regulating AKT signalling has been suggested by several studies in the setting of cancer[24,53,54]. Our work further supports the importance of this functional interaction in pathological conditions in vivo, although a key difference is that we demonstrate regulation of T308 phosphorylation and not S473 in early disease, building on emerging literature which suggests an important functional role for differential phosphorylation of these residues[55]. The precise mechanism by which this occurs remains to be established, but phosphorylation at residue T308 is ordinarily facilitated by the actions of PDK1 which we show

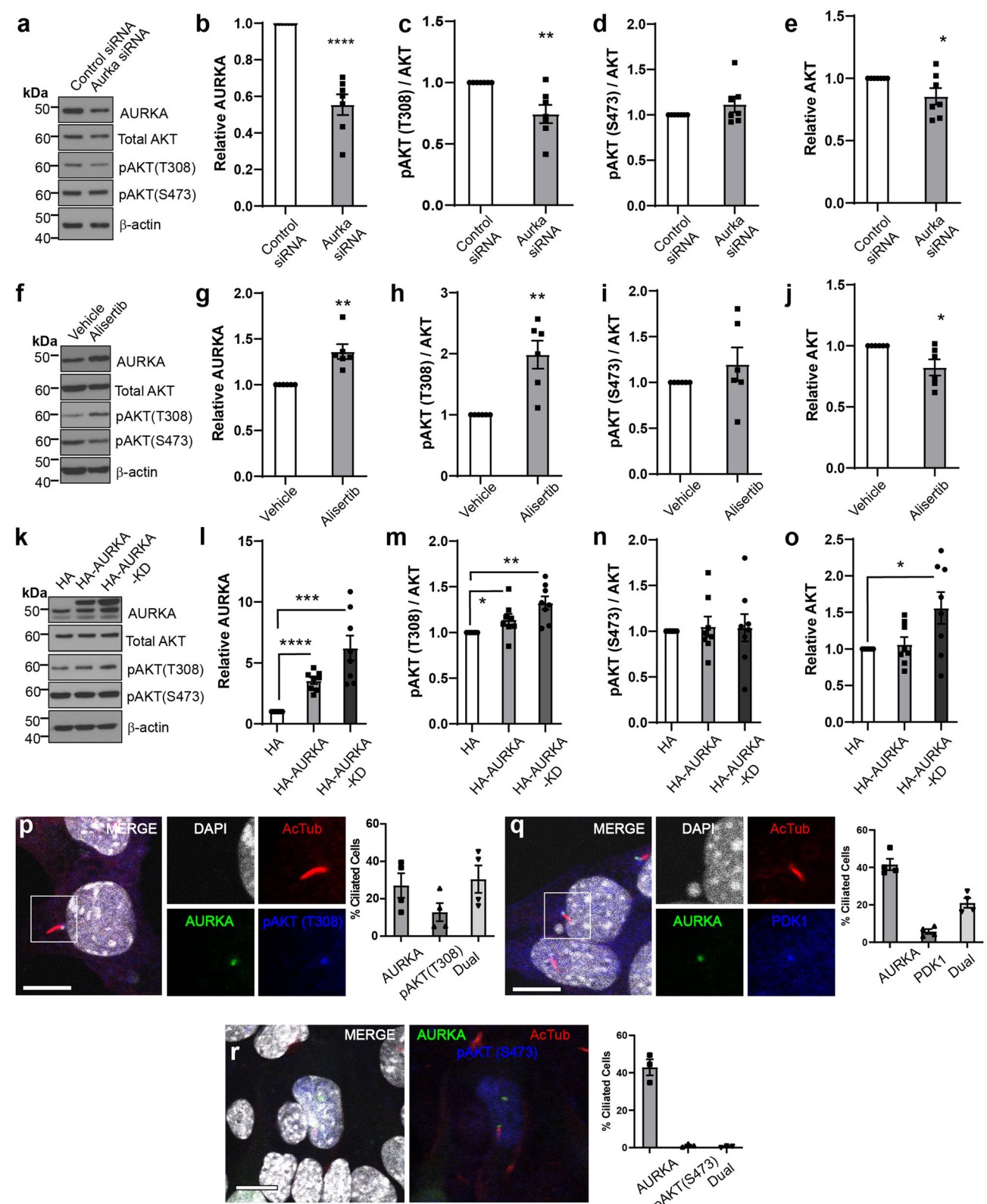

co-localises with AURKA and AKT at the basal body. PIP3 is known to recruit AKT to membranes for subsequent PDK1-mediated phosphorylation[56] and it may be that AURKA is involved in an analogous manner to scaffold interactions at the primary cilia.

We find that AKT inhibition results in significant reductions in cyst number and size in models of both ADPKD and JS, although the magnitude of improvement did not match *Aurka* deletion. This may be due

to differences in timing, where *Aurka* deletion occurred embryonically whilst MK2206 treatment started at P4 when cysts were already present. However it also suggests that AURKA may act through additional pathways other than AKT. AKT inhibitors have historically have been trialled in terminal cancer patients due to their potential for systemic toxicity, so AURKA based therapeutics (with comparatively less expression in adult tissues), may represent a more attractive clinical

**Fig. 5 | AURKA regulates pAKT (T308) status and co-localise. a–e** Western blots and densitometry of mIMCD3 cell lysates transfected with control siRNA or *Aurka* siRNA#1 (cultured for 24 h). *P* values from left to right; *p* = 0.0001, *p* = 0.0069, *p* = 0.0360. **f–j** Western blots and densitometry of mIMCD3 cell lysates treated with Alisertib for 48 h under growth conditions and probed as indicated. *P* values from left to right; *p* = 0.0033, *p* = 0.0039, *p* = 0.0223. **k–o** Western blots and densitometry of mIMCD3 cell lysates transfected with HA, HA-AURKA or HA-AURKA KD expression plasmids (cultured 24 h) and probed as indicated. Densitometry values are sum of multiple bands observed. *P* values from left to right for pairs indicated; *p* = 1.8 × 10⁻⁵, *p* = 0.0007, *p* = 0.0313, *p* = 0.0014, *p* = 0.019. **p** Immunostaining for DAPI, AURKA, and pAKT (T308) in mIMCD3 cells showing co-localisation at the basal body and the distribution of protein localisation. **q** Immunostaining for DAPI, AURKA, and PDK1 in mIMCD3 cells showing co-localisation at the basal body and the distribution of protein localisation (bar = 10 μm). **r** Immunostaining for DAPI, AURKA, and pAKT (S473) in mIMCD3 cells showing no co-localisation at the basal body (bar = 10 μm). All graph data indicate mean ± S.E.M. Control references defined as 1. pT308 pAKT (T308). **a–e** *n* = 7, **f–j** *n* = 6, **k–o** *n* = 8, **p–q** *n* = 4, **r** *n* = 3. Additional data in Figs. S6 and S7. Western blots represent biological replicates with each independent experiment set run on a separate gel or two replicates on a gel. Each control sample within a replicate set was then defined as 1. The western membranes were cut to 40–60 kDa interval before antibody probing, repeated stripping and reprobing of the same membrane to generate datasets. Individual western blot scans, *n* and data point values provided in supplementary material-Source Data File. All graph data indicate mean ± S.E.M.

approach. Inhibition of mTORC1 signalling downstream of AKT using Rapamycin analogues showed considerable early promise as potential ADPKD therapies. However, their use in PKD mouse models reduced cyst size but not number[14,57] and clinical trials in ADPKD patients showed minimal to no beneficial effects[58,59]. The studies presented here indicate that AKT may instead act on other downstream targets besides mTORC1 that contribute to cyst formation, an observation consistent with our previous findings[43]. As well as preventing cyst formation, our experiments examining *Aurka* deletion in existing cysts suggests an ongoing role for AURKA in cyst growth.

Deletion of *Aurka* in the developing renal collecting duct does not substantively alter the normal arborisation of the ureteric bud or the maintenance of this tissue after birth. This suggests that AURKA is dispensable in this tissue, which is surprising given its apparently obligate roles in cell division[18,19,27,28] and early embryonic development[21]. Under normal conditions, we detect little expression of AURKA in the renal tubules and it may be that other factors can compensate for its actions. Our work showing that AURKA is both dispensable to normal kidney homoeostasis and pathologically up-regulated in disease identifies AURKA an attractive therapeutic target for ameliorating PKD development.

It is notable that Alisertib treatment of *Inpp5e*^Δ/Δ mice does not phenocopy *Aurka* genetic deletion; nor does drug treatment of cultured cells match *Aurka* siRNA knockdown or mimic the effect of over-expression of AURKA kinase dead mutants. Instead, we find that Alisertib promotes AURKA accumulation, leading to rebound AKT activation in a kinase-independent manner. This observation provides an explanation for why cyst number increases in *Inpp5e*^Δ/Δ mice treated with Alisertib compared to those in which *Aurka* is simultaneously deleted. Although the function of AURKA in cilia disassembly and mitosis is thought to relate to its kinase activity[60], kinase-independent functions of the protein have also been identified[61]. Significantly, we observed that inhibition of AURKA with Alisertib can still trigger cilia resorption and cell proliferation in a process associated with activation of AKT. These findings indicate that caution should be exercised when inferring AURKA function using Alisertib, because although the drug acts as an inhibitor of kinase activity, it also has the potential to act as an agonist of the protein's kinase-independent functions. These agonist properties such as increasing total AURKA levels and increasing AKT pT308 ratios were also observed following treatment of cells with MK5108, VX680 and C1368 (Fig. S6a–c), suggesting this may be shared feature of ATP competitive AURKA kinase inhibitors. Such properties could suggest an explanation for the poor clinical trial results for Alisertib, MK5108 and VX680, indicating that alternative means to block AURKA's functions may be needed.

Our analysis now positions AURKA as a major upstream regulator of AKT which, in addition to being a transcriptional target[15], creates a feed-forward loop which strengthens AURKA driven AKT activation - amplifying the impact of *Inpp5e* or *Pkd1* loss. However, we cannot exclude roles for intermediary AURKA binding partners in regulating these changes. Moreover, while the co-localisation of these proteins in the primary cilium suggest that this is one subcellular compartment critical for these interactions[62,63], we cannot exclude functionally important roles at other subcellular locales. For example, both proteins are also localised in mitotic spindles and mitochondria[49,64], the function of the latter being impaired in ADPKD[65,66]. Given the pleiotropic role that AURKA plays in many aspects of cellular homoeostasis the protein may also regulate cyst development via pathways other than AKT. Better understanding the direct and indirect mechanistic effects of AURKA on these cyst promoting pathways will be critical to understanding it's central role in disease progression. Nonetheless, the identification of a shared AURKA-AKT-mediated cyst promoting axis which underpins disease development in different types of PKD provides important molecular insights into renal cyst formation. These findings also identify a pathway which may be amenable to therapeutic interventions aimed at preventing the development of kidney cysts.

## Methods

### Study approval

Animals were housed, welfare monitored, used for experiments and euthanised under the approval of the Monash University Animal Ethics Committee (MARP1) in a manner compliant with the guidelines of the Australian Bureau of Animal Welfare. This study follows ARRIVE guidelines.

### Antibodies

See Supplementary Table 1: List of Antibodies, Lectins and Stains

### Animals

*Inpp5e*^tm1Cmit C57BL6J (MGI:5823279) mice have been described previously[13,14]. *Aurka*^tm1.1Tvd C57BL6J mice[21] were imported from Jackson Laboratory (USA, Stock No: 017729, RRID:IMSR_JAX:017729). *Tg(Hoxb7-cre)*^13Amc *(Hoxb7-cre)* mice[34] were kindly provided by the McMahon lab (University of Southern California, USA). *Hoxb7-cre* mice were predominantly C57BL6J with very minor (<5%) Swiss Weber contribution (RRID:IMSR_JAX:004692). *Pkd1*^tm2Ggg C57BL6J[67] were sourced from the Jackson Laboratory (USA, Stock No: 010671, RRID:IMSR_JAX:010671). Adult-onset ADPKD mice were provided by Prof Stefan Somlo[17] and the floxed *Pkd1*^tm1Som allele was replaced with the floxed *Pkd1*^tm2Ggg (detailed above). The *Pax8*^rtTA and *TetO-Cre* mouse lines have been previously described[17] (B6.Cg-Tg(Pax8^rtTA2S*M2)1Koes/J Tg(TetO-Cre)^1Jaw/J, RRID:IMSR_APB:8010). *Pkd1 RC* mice (*Pkd1*^tm1.1Pcha, RRID:MGI:5476836)[45] on a C57BL6J background were a gift from Prof Gopi Rangan. Gene deletion of *Pkd1* and/or *Aurka* was triggered by doxycycline administration in drinking water at P28 as previously described[17]. *AURKA-V5* mice were developed by the CRISPR mediated insertion of a 5'GGAAAGCCCATTCCCAACCCACTTTTGGGCTTGGACAGTACT oligo encoding a codon optimised Simian Virus 5 (V5) tag sequence immediately before the STOP codon (Australian Phenomics Network (APN) and Monash Genomic Modification Platform (MGMP)). Mice were crossed at the Monash University Research Platform - Animal Research Laboratory (MARP-ARL). Animals of both sexes were used in cohorts as most models did not show sexual dimorphism, with the exception of *Pkd1 RC* mice. Sex was not recorded for neonatal

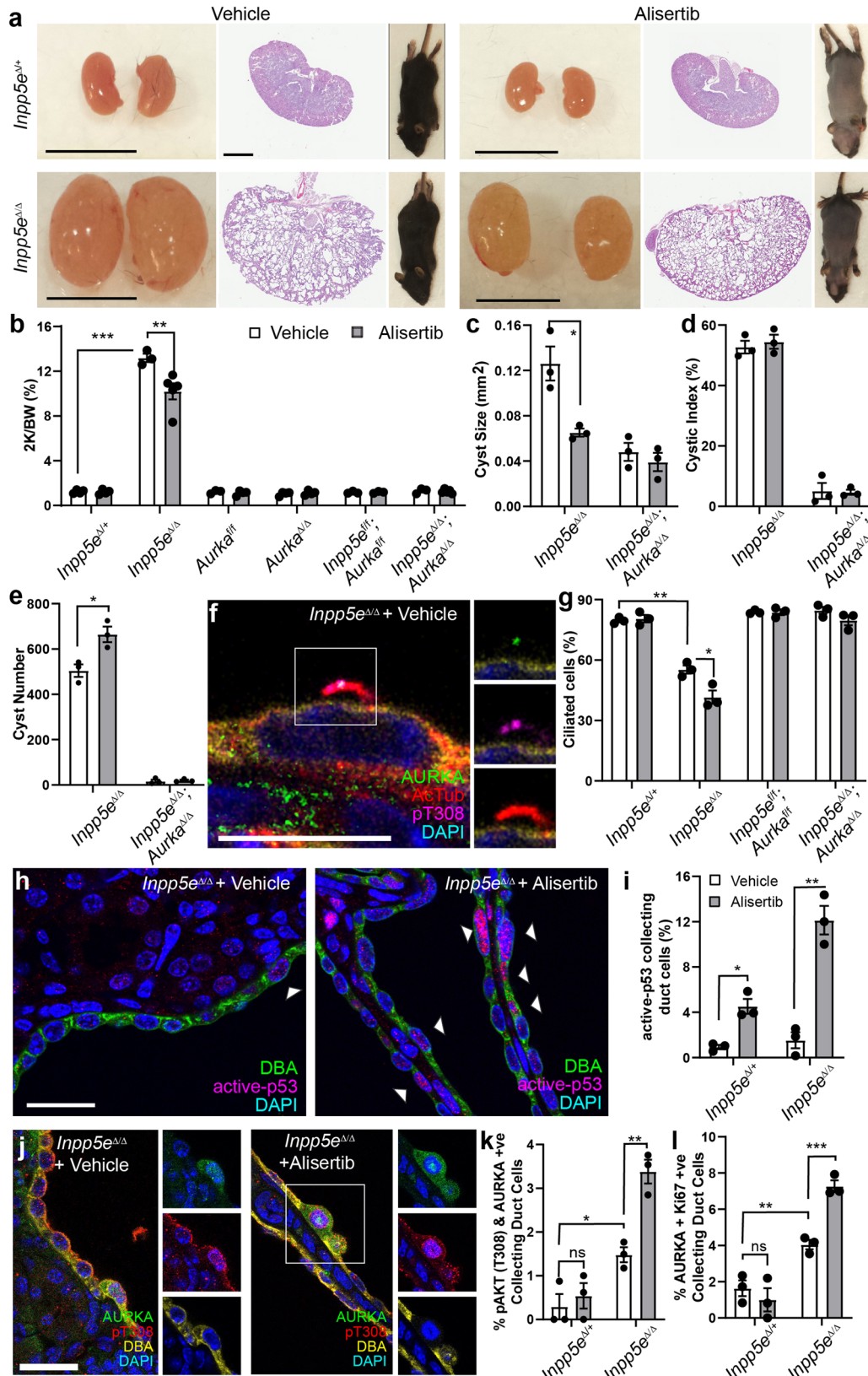

studies, however animal numbers split by sex are given for adult mouse studies in the Source Data File. In the case of Pkd1 RC studies, sexes were approximately 50:50 balanced overall (see Source Data File for exact sex composition) and used between P28 and 6 months of age before significant sex divergence occurs (Fig. S4a). *Aurka* was deleted

from RC models using the *Pax8-rtTA TetO-Cre* system outlined above. Time points for analysis are as indicated for P0, P4, P11, however P21 refers to range of P17–P21, P60 refers to the range P60-P75 and P150 refers to the range P150-P165. 12-weeks refers to P84, 18-weeks refers to P126, (except one *Pkd1*$^{\Delta/\Delta}$; P124), and 20 weeks refers to P140 (except

**Fig. 6 | Alisertib increases cyst number in vivo in JS model. a** Mice, whole kidneys and sections following treatment with vehicle or Alisertib (bar = 1 cm for kidneys; 2 mm for sections). **b–e** Quantification of combined kidney weight over total body weight, cyst number, cyst size and cystic index for vehicle and Alisertib-treated mice. *P* values are 0.0003 and 0.0059 (**b**), 0.0247 (**c**) and 0.0117 (**e**). **f** Ciliary co-localisation of AURKA and pAKT (T308) in *Inpp5e*$^{\Delta/\Delta}$ kidneys (bar = 10 μm). **g** Quantification of ciliated collecting duct cells following treatment with vehicle and Alisertib. *P* values from left to right; *p* = 0.0012, *p* = 0.0162. **h**, **i** Active-p53 in collecting ducts of *Inpp5e*$^{\Delta/\Delta}$ kidneys (arrow = nuclear p53; bar = 25 μm). *P* values

from left to right; *p* = 0.0123, *p* = 0.0022. **j**, **k** Staining and quantification of AURKA and pAKT (T308) +ve collecting duct cells. *P* values from left to right; *p* = 0.0173, *p* = 0.0035. **l** Quantification of the proportion of dual AURKA/Ki67+ve collecting duct cells. *P* values from left to right; *p* = 0.0072, *p* = 0.0009. All graph data indicate mean ± S.E.M. pT308 pAKT (T308). ns not significant. White bars = vehicle; Grey bars = Alisertib treatment. **a**–**b** *n* = 3–5, **c**–**l** *n* = 3. All data from P15. Additional data in Fig. S8. Exact *n* and data point values provided in supplementary material- Source Data File.

---

one *Pkd1*$^{\Delta/\Delta}$; P135). Animals were fed Barastoc mouse breeder irradiated cubes (RID22470) with autoclaved water, and housed on sawdust bedding in Optimice cages at 18–24 °C 40–70% humidity and 12 h light/dark cycle.

### Animal genotyping
Genotyping for *Inpp5e* wt, floxed, *Aurka* wt, floxed, and Δ alleles was performed as described previously[14,21]. *Pkd1* wt, floxed and KO alleles genotyping was as outlined by Jackson Laboratories[67]. Genotyping of *cre* was performed using the Jackson Laboratory master generic *cre* protocol. *Pkd1 RC*, *Pax8-rtTA* and *TetO-Cre* genotyping was performed with the services of Transnetyx or completed in house with primers 5′ CCATGTCTAGACTGGACAAGA and 3′CTCCAGGCCACATATGATTAG for *Pax8*$^{rtTA}$ and 5′GCAGAGCTCGTTTAGTGAAC and 3′TCGACCAGTT-TAGTTACCC for *TetO-Cre* provided by Prof Stefan Somlo. All PCRs used GoTaq green master mix (Promega). Genotyping was also performed with the services of Transnetyx. *AURKA-V5* PCRs were developed in house with primers 5′CTGCCGAGGGCTGGTGTTTT and 3′ ACAAGGACACCTTGGGGCAC. Cycles were 1× denaturing 94 °C for 2 min, 35× denaturing 94 °C for 20 s + annealing 64 °C for 20 s + extension 72 °C for 30 s, and polishing 72 °C for 5 min. WT PCR bands were 495 bp while V5 incorporated bands ran higher at 537 bp.

### Mouse Alisertib treatments
Alisertib treatment was performed based on previously published protocols[32,33]. Starting at P9 *Inpp5e*$^{\Delta/\Delta}$ and *Inpp5e*$^{\Delta/+}$ mice were treated once daily for 5 days with Alisertib delivered orally by pipette in a suspended solution with a final composition of 3.6 mg/ml Alisertib (S1133, Selleck Chemicals) in 10% 2-hydroxypropyl-β-cyclodextrin (H5784-10ML, Sigma–Aldrich) with 1% (vol/vol) sodium bicarbonate (S8761-100ML, Sigma–Aldrich) and 5% glucose (49163-100ML, Sigma–Aldrich). Mice were dosed with a volume to deliver Alisertib at 10 mg/kg. Mouse body weights were recorded daily. Alisertib was largely insoluble and vortexed before each use to resuspend. Mice were then given 2 recovery days before sacrifice at P15. *Aurka*$^{\Delta/\Delta}$ and *Inpp5e*$^{\Delta/\Delta}$;*Aurka*$^{\Delta/\Delta}$ treatment groups were also included to determine the specificity of any phenotypes observed, given Alisertib's broad inhibition profile[31,68]. Solutions were stored at 4 degrees Celsius for up to 1 month.

### Mouse MK2206 treatments
Vehicle solution was made up of 30% Capsitol and 5% glucose while MK2206 solution was made up of 30% Capsitol (Cydex, RC-0C7-020), 5% glucose (49163-100ML, Sigma–Aldrich) and the indicated concentration of MK2206.2HCl (S1078, Selleck Chemicals): 3.6 mg/ml MK2206.2HCl for 10 mg/kg dosage, 13.5 mg/ml MK2206.2HCl for 37.5 mg/kg dosage and 27 mg/ml MK2206.2HCl for 75 mg/kg dosage. Solutions were delivered orally by pipette and stored at 4 degrees Celsius for up to 2 weeks.

Treatment of *Inpp5e*$^{\Delta/\Delta}$ and *Inpp5e*$^{\Delta/+}$ mice commenced at P9 and was performed daily up to P14 before sacrifice at P15. Mice were dosed with a volume to deliver MK2206 at 75 mg/kg and body weights recorded daily. *Pkd1*$^{\Delta/+}$ and *Pkd1*$^{\Delta/\Delta}$ were treated once daily for 7 days (from P4 until P10) before sacrifice at P11. Mouse body weights were recorded daily to determine the

volume of vehicle or MK2206 solution at 10 mg/kg, 37.5 mg/kg or 75 mg/kg given.

### Embryonic tissue collection & OPT analysis
Embryos were collected at E14.5 following ethical guidelines and kidneys analysed by optical projection tomography as previously described[69].

### Postnatal tissue collection
Mice were culled via cervical dislocation or decapitation, as most appropriate for their age following ethical guidelines. Mice were measured for body weight, with ear tissue, urine and kidneys collected. Ear tissue and urine was stored at −20 degrees. Newborn and P4 kidneys were fixed whole in 4% PFA for 16 h at 4 degrees, while the kidney capsule membrane was removed from older mouse kidneys, with the kidney cut into halves and fixed in 10% NBF for 16 h at room temperature. Half the kidney samples were processed for histology and the other reserved for RNA preparations and/or western blotting.

### Kidney function analysis
Urine Albumin Creatinine Ratios (ACR) were performed with Albumin mouse ELISA kit (ab108792, Abcam) and Creatinine assay kit (ab65340, Abcam) according to manufacturer's protocol. Blood serum Urea Nitrogen (BUN), and serum Creatinine values were determined using an ARL Analyzer (Dupont, North Ryde, NSW, Australia) with the services of Prof. David Nikolic-Paterson (Monash Health).

### Histological analysis and microscopy
Kidney tissues were paraffin-embedded and sectioned at 4 μm for histology and at 10 μm for cilia analysis. Antigen retrieval was performed in Citrate buffer pH6 using Tefal pressure cooker or using a DAKO PTlink system. Antibody staining was performed as described elsewhere[70], except triton x-100 was included in blocking buffers and coverslips were mounted using Prolong Gold (Invitrogen). Antibodies and stains are listed below. Imaging was performed using an Aperio brightfield and fluorescent scanners, Olympus Fluoview 500, Nikon C2, Leica DMI8 or Leica SP8 confocal microscopes.

### Cell lines
mIMCD3 cells (ATCC CRL-2123, RRID:CVCL_0429) were grown as previously described[15]. For western blot experiments, cells were seeded in 6 well dishes at $1.3 \times 10^5$ cells per well in growth media and incubated overnight at 37 °C in 5% CO2. For serum-starvation conditions, 0.5% serum containing was placed on mIMCD3 cells and wells harvested after 24 h. For serum-re-stimulation conditions, mIMCD3 cells were first placed in 0.5% serum media for 24 h before addition of fresh growth media before harvest. For immunofluorescence experiments, mIMCD3 cells were grown on collagen I-coated 22 × 22 mm coverslips, seeded at $1.3 \times 10^6$ cells per well, in Glutamax containing growth media for high-density adherence overnight. Cells were then switched to a media without serum or L-glut/Glutamax for 48 h to encourage ciliation. Coverslips and attached cells were processed as described in[71]. Drug treatments included 1μM Alisertib (S1133, Selleck Chemicals), 1 μM MK5108 (S2770, Selleck Chemicals), 2 μM Cyclopropane carboxylic acid (C1368, Sigma–Aldrich), 0.5 μM VX680 (SML2158, Sigma–Aldrich) all in DMSO.

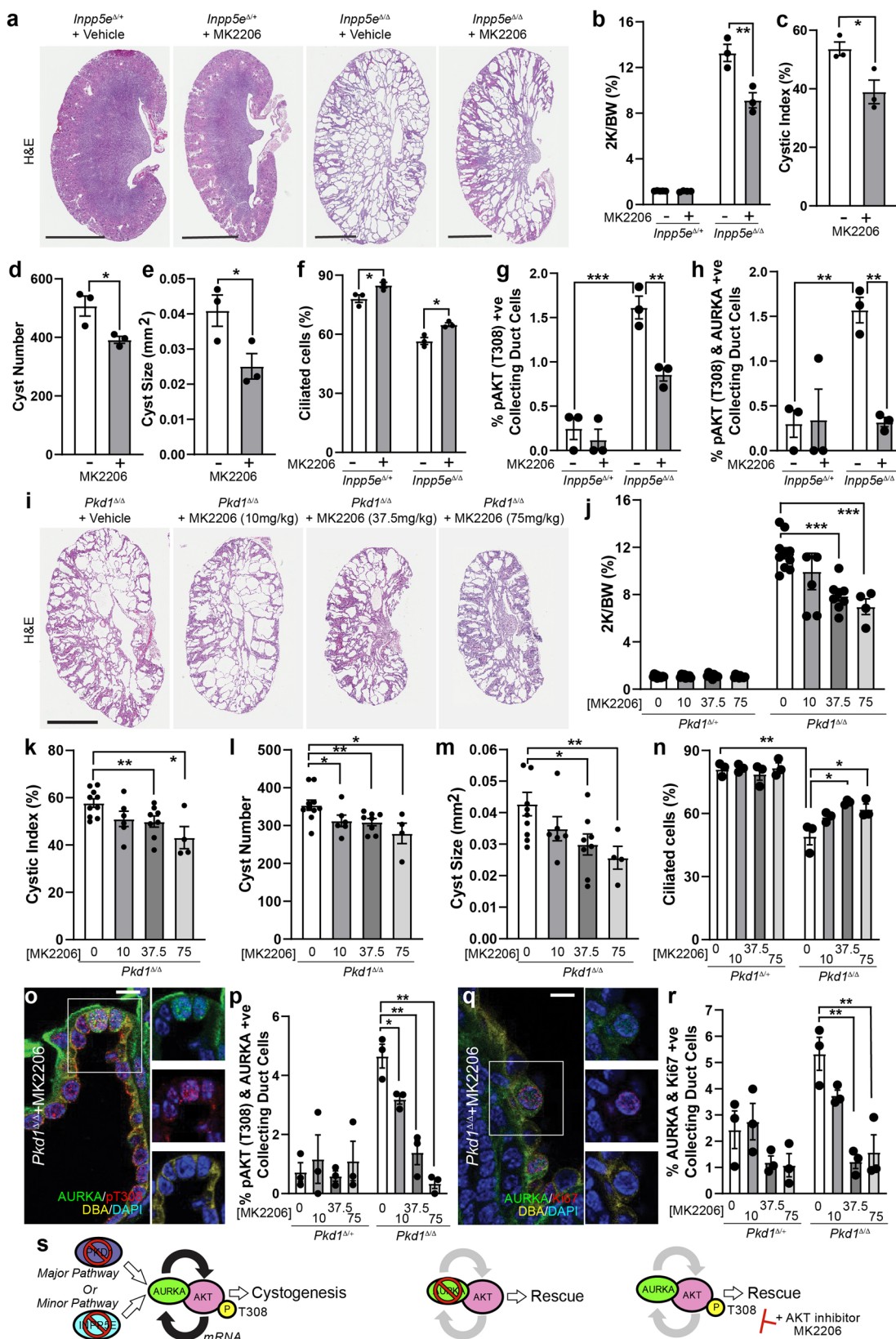

## Transfection

Lipofectamine 3000 Reagent kit (Invitrogen) was used for plasmid transfection of mIMCD3 cells according to manufacturer's protocol (3.75 µl of Lipofectamine 3000, 2.5 µg DNA and 5 µl of P3000 reagent in each well). siRNA transfections were also performed with Lipofectamine 3000 (Invitrogen) according to manufacturer's protocol

(0.75 µl of Lipofectamine 3000, 25 pmol siRNA in each well). Plasmids included pCGN (HA-tag, RRID:Addgene_53395), pCGN-HA-AURKA (gift from Olga Plotnikova), and pCGN-HA-AURKA KD (made using Q5 Site-Directed Mutagenesis Kit NEB.E0554S according to manufacturer's protocol to contain the K162R variant described for pcDNA3.1-mRFP-AURKA K162R[15]). All plasmids were prepared with

**Fig. 7 | Inhibition of AKT constrains cyst formation. a, i** Whole kidneys from mice treated with vehicle or MK2206 (bar = 1 cm (*Inpp5e*); 0.5 cm (*Pkd1*)). **b, j** Quantification of the combined kidney weight over total body weight percentage (2 K/BW%). *P* values as indicated are $p = 0.0075$ (**b**), $p = 1.8 \times 10^{-5}$ for 37.5 mg/kg; $p = 0.0006$ for 75 mg/kg (**j**). **c, k** Cystic index quantification, $p = 0.0233$ (**c**), $p = 0.0096$ for 37.5 mg/kg; $p = 0.0226$ for 75 mg/kg (**k**). **d, l** Quantification of average cyst number per section, $p = 0.0323$ (**d**), $p = 0.0314$ for 10 mg/kg; $p = 0.0080$ for 37.5 mg/kg; $p = 0.0306$ for 75 mg/kg (**n**). **e, m** Average cyst cross-sectional size, $p = 0.0267$ (**e**), $p = 0.0102$ for 37.5 mg/kg; $p = 0.0046$ for 75 mg/kg (**m**). **f, n** Ciliation, $p = 0.0249$ and 0.0118 for (**f**) and $p = 0.0035$, $p = 0.0269$, $p = 0.0327$ for (**n**). **o** Staining for AURKA, DBA and pAKT (T308) in mice exposed to MK2206 or vehicle controls (bar = 10 μm). **g** Quantification of pAKT (T308) +ve co-labelled collecting duct cells. *P* values from left to right; $p = 0.0008$, $p = 0.0062$. **h, p** Quantification of pAKT (T308)/AURKA +ve co-labelled collecting duct cells.

*P* values from left to right; $p = 0.0018$, $p = 0.0034$ for (**h**) and $p = 0.0263$, 0.0024 and 0.0017 for (**p**). **q, r** Staining and quantification of Ki67/AURKA collecting duct cells. *P* values from left to right; $p = 0.0061$ and 0.0075, bar = 10 μm. All graph data indicate mean ± S.E.M. pT308 pAKT (T308). White bars = vehicle; Grey bars = MK2206 treatment. **a–h** $n = 3–5$, **i–j** $n = 4–12$, **k–m** 4–10, **n–r** $n = 3$. *Inpp5e* model used MK2206 at 75 mg/kg and data is at P15. *Pkd1* model used MK2206 doses as indicated and data is at P11. Additional data in Fig. S8. **s** A model of the role of AURKA and AKT in regulating cystogenesis in PKD. Deletion of either *Pkd1* (Major pathway driver) or *Inpp5e* (Minor pathway driver) result in upregulation of AURKA and enhanced AKT activity through phosphorylation of T308 to form a self-reinforcing signalling loop driving cystogenesis. Co-deletion of *Aurka* prevents AKT activation and stops cyst formation. Inhibiting AKT activity with MK2206 breaks the feedback loop and also reduces cyst formation. Exact *n* and data point values provided in Source Data File.

Qiagen plasmid midiprep kits. siRNAs were All Stars negative control siRNA (SI03650318, Qiagen) and Mm_Aurka_1 Flexitube siRNA (SI00908803, Qiagen). Alisertib (10 mM in DMSO, S1133, SelleckChem) was added to media at 1 μM final concentration where indicated. 16 h after plasmid and siRNA transfections, fresh growth media was placed on mIMCD3 cells and wells harvested up to 24 h later.

## AlphaLISA protein analysis
AlphaLISA Surefire Ultra Kits (#ALSU-CUSTOM, Perkin Elmer/TGR Bioscience) were a gift from Perkin Elmer. Kits contained antibodies to detect murine p4EBP1 (T37/46), pAKT (T308) pAKT (S473), AKT1 and GAPDH. Kidney tissue sampled were lysed for 45 min rocking at 4 degrees in AlphaLISA Surefire Ultra lysis buffer. Protein extracts were assayed according to manufacturer's protocol.

## Immunoblotting
Tissue extracts were prepared in AlphaLISA Surefire Ultra lysis buffer or 0.25% NP40 in Tris-buffered saline pH 7.4 with Roche complete mini protease and PhosSTOP inhibitor tablets. Cell extracts were prepared in 1% Triton X-100 in Tris-buffered saline pH 7.4 with Roche complete mini protease and PhosSTOP inhibitor tablets. Immunoblotting was performed using the antibodies in the Supplementary table 1: List of Antibodies, Lectins and Stains. Western membranes were run with biological samples, probed and re-probed (following stripping when required) with the collection of antibodies indicated to acquire datasets. For some experiments biological samples were run in parallel on duplicate membranes and antibodies probed across the Gel set to acquire datasets due to limitations of repeated stripping and reprobing. Densitometry values were determined using ImageJ. Failed western blots attempts were excluded. Figure S5g has an outlier omitted as outlined in figure legend and Source Data file.

## Co-Immunoprecipitation
Two kidneys at P15 (*Inpp5e*$^{\Delta/\Delta}$ and controls) or P11 (*Pkd1*$^{\Delta/\Delta}$ and controls) were homogenised in 3 ml of 0.25% NP40 in Tris-buffered saline pH 7.4 with Roche complete mini protease and PhosSTOP inhibitor tablets and extracted for 2 h rocking at 4 degrees. Alternatively, one testis was homogenised in 4 ml of 0.25% NP40 in Tris-buffered saline pH 7.4 with Roche complete mini protease and PhosSTOP inhibitor tablets and extracted for 2 h rocking at 4 degrees. The supernatant fraction was collected and precleared with 240 μl of Pierce Protein A/G agarose bead slurry (Thermo Fisher Scientific, LTS20421) for 2 h rocking at 4 degrees. The solution was passed through a Pierce spin filter column (Life Technologies, 69725) and gravity-fed flow through fraction collected. 30 μl of goat anti-V5 agarose beads (Abcam, ab1229) was added to 1.25 ml lots of precleared extract and incubated overnight at 4 degrees. The beads were collected by passing through a Pierce spin filter column (Life Technologies, 69725) on a vacuum manifold and washed with 1 ml cold extraction buffer 6 times, before reducing

buffer was added to samples, boiled, eluted, stored frozen then immunoblotted.

## RNA sequencing analysis
Snap-frozen P4 kidney tissues were crushed using micropestles in RLT Buffer and RNA was extracted from these tissues using QiaShredders followed by Qiagen RNAeasy Mini Kits according to manufacturer's protocol. RNA concentration and integrity were determined by Aligent bioanalyzer. cDNA library preparation was carried out using NEB Next Ultra Directional RNA Library Prep Kit for Illumina RNA sequencing (E7420S) and sequenced on an Illumina HiSeqX10 machine in 150 bp paired end format. Sequence data was processed with Skewer adaptor trimmer and mapped with HiSat2 to the *Mus musculus* GRCm38_v90 genome assembly. The resulting ordered Bam files were analysed in Seqmonk v1.43 software using default settings. Samples were grouped as *Inpp5e* KO (*Inpp5e*$^{\Delta/\Delta}$ mice), *Pkd1* KO (*Pkd1*$^{\Delta/\Delta}$ mice), *Aurka* KO (*Aurka*$^{\Delta/\Delta}$ mice), *Inpp5e* double KO (*Inpp5e*$^{\Delta/\Delta}$; *Aurka*$^{\Delta/\Delta}$ mice), *Pkd1* double KO (*Pkd1*$^{\Delta/\Delta}$; *Aurka*$^{\Delta/\Delta}$ mice) or Control Cre (*Hoxb7-cre* mice). Bam files are available at https://doi.org/10.26180/24521446.v2 or https://bridges.monash.edu/articles/dataset/Bam_and_BAI_files/24521446. A minimum of 13 million reads was obtained per sample and library duplication and QC metrics were assessed. Differentially expressed genes between groups were determined using the count-based DeSeq2 method with a multiple testing corrected *p* value cut off of $p < 0.05$ and independent filtering applied. This list was filtered following count/total sequences log2 transformation for fold changes of ±1.5. Significant pathway changes were analysed in the filtered gene list using online tool String-db.org. Venn diagrams were created using Venny 2.1 (https://csbg.cnb.csic.es/BioinfoGP/venny.html).

## Quantification
Values in all figures are presented as Mean ± Standard Error of the Mean (S.E.M) from biological replicates. 2 K/BW% was calculated as the combined kidney weight over total body weight percentage. Cyst identity was measured as a proportion of cyst epithelial area for each marker out of total sum of cyst epithelial areas of all markers (adjusted for marker co-localisation). Cystic index % is calculated as the proportion of cystic space occupied out of total kidney area in stained kidney longitudinal cross-sections. Cyst number reflects the average number of cysts counted per stained longitudinal kidney cross-section. Cyst size is the average cross-sectional cyst size (mm²) of cysts sampled from stained kidney longitudinal cross-sections. Staining includes H&E for all cyst types, or tubule-specific cyst subtypes with DBA, LTL or UMOD staining. Cyst index, number and size calculations were performed in a semi-automated process throughout with the exception of *Pkd1 RC* mouse studies utilising an Artificial Intelligence-based pipeline described in the next section. Cilia length was analysed from confocal z-stack images, where at least 10 cilia were measured per image, with 10 images taken per mouse, with 3 mice analysed per genotype giving a total of at least 300 cilia measurements per condition. For in vivo cell

proportions, at least 10 random fields across the kidney section were imaged and the number of positive cells for a given factor (Cilated, Ki67+ve, γ-H2AX +ve, AURKA+ve, AKT pT308 +ve, p53+ve etc) counted and divided by the number of collecting duct cells determined (across all images) using ImageJ. Due to escape of *cre* activity at the *Aurka* locus in cysts, unless otherwise indicated references to *Inpp5e*^Δ/Δ;*Aurka*^Δ/Δ immunostains and their quantifications reflect non-cystic regions throughout this study. n values reflect number of mice or independent biological samples for mIMCD3 cell cultures, thus defining the number of experimental replicates.

## Pkd1 RC mouse AI quantification

Longitudinal sections at the approximate mid-line of the kidney containing medulla and cortex were histologically processed and haemotoxylin/eosin stained. Sections found not to be at the approximate mid-line and/or exhibiting damage were excluded from analysis. After scanning with an Aperio scanning microscope (Leica Microsystems), tiff images were exported from the ImageScope digital slide files then batch processed using Image J script to subsample, remove background signal and imaging artefacts. Ilastik[72] pixel classification was then used to delineate tissue, cyst and slide background. AI object classification was trained on and compared with manually analysed data subset before attempting full cohort analysis. Measurements for each classified object were exported and processed using an R script to perform statistical analysis and validity tests.

## Statistics & reproducibility

Data sample sizes were determined from prior experience. Animals (or their entire litters in the case of neonates) were randomly assigned to drug treatments. Investigators were not blinded to allocation during experiments and outcome of assessment. Rare animals with hydronephrosis, preventing analysis of a matched kidney pair, were excluded from analysis. No remaining animals were intentionally excluded barring sample collection or processing failures. For some experiments, a subset of animals were randomly selected for analysis from within a larger cohort. In general, datasets were analysed using the unpaired Student's *t*-test assuming unequal variance (Welch's *t*-test) with a limited number of comparisons precluding multiple testing correction, unless otherwise indicated, in Microsoft Excel or GraphPad Prism version 9.5.1, where $P \leq 0.05$ values were considered significant. One-tailed tests were selected as appropriate for hypothesis. A Mantel-Cox test was performed to assess survival. *P* values indicated on figures follow the ****$p \leq 0.0001$, ***$p \leq 0.001$, **$p \leq 0.01$, *$p \leq 0.05$, #$p \leq 0.06$, NS Not Significant convention. Actual *p* values are given in figure legends.

## Reporting summary

Further information on research design is available in the Nature Portfolio Reporting Summary linked to this article.

# Data availability

Data supporting the findings of this study are available within the paper, Supplementary Information and Source Data File. Raw RNAseq data is available at https://doi.org/10.26180/24521446.v2 or https://bridges.monash.edu/articles/dataset/Bam_and_BAI_files/24521446. Requests for resources and reagents should be directed to and will be fulfilled by the lead contact, Prof Ian M. Smyth (ian.smyth@monash.edu). Source data are provided with this paper.

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

## Acknowledgements

The authors would like to acknowledge the assistance of the Monash Histology Platform (MHP), Australian Phenomics Network (APN), Monash Genomic Modification Platform (MGMP), Monash Micro Imaging (MMI), the Biochemistry Imaging Suite manager Irene Hatzinisiriou and the Monash Animal Research Platform - Animal Research Laboratory (MARP-ARL) - in particular animal technicians Joel Eliades, Kelly Sperne and Samantha O'Dea. We would also like to thank Olga Plotnikova and Anna Nikonova for advice regarding Alisertib animal experiments and the former for providing the pCGN-HA-AURKA plasmid. We also thank Ash Spencer, Tin Wong, Melissa Remy, Manizha Shekibi, and Elisha Papadimitriou for technical assistance. This work was supported by grants from the Australian National Health and Medical Research (NHMRC) APP1046174 and APP1182539. This research was funded by a grant from the Polycystic Kidney Disease Foundation, pkdcure.org. The Foundation had no role in study design, data collection and interpretation, or the decision to submit the work for publication. MST was supported by a Monash International PhD student scholarship. AKZ was supported by an Australian Government Research Training Program (RTP) Scholarship. IMS was supported by a Senior Research Fellowship of the NHMRC (APP1106516).

## Author contributions

Co-first authors, M.S.T. and A.K.Z. performed data acquisition and analysis. M.S.T. co-wrote the manuscript. Co-first, co-supervising and co-corresponding author, D.L.C. co-directed the project, performed data acquisition, analysis and co-wrote the manuscript. Co-first and co-corresponding author order defined by mutual agreement. K.M.S. performed data acquisition and analysis. L.K.J. contributed to data acquisition. S.E.C., J.M.D. & C.A.M. provided materials and intellectual input. I.M.S. conceived the project, supervised, mentored, co-directed and co-wrote the manuscript. All authors assisted with manuscript editing.

## Competing interests

I.M.S., D.L.C. and M.S.T. are inventors on patent application WO2021081580A1 and I.M.S. and D.L.C. are co-founders of xCystence Bio, based on this work. Otherwise, the authors have declared that no conflict of interest exists.
