## [Peer Review File · Nature Communications]

REVIEWER COMMENTS

Reviewer #1 (Remarks to the Author):

Introduction

In this manuscript, the authors seek to establish a link between Aurora A Kinase (AURKA), AKT, and Polycystic Kidney Disease (PKD). They present a highly convincing case that activity of this kinase is instrumental in the development of this devastating disease and its action is regulated by AKT not mTOR. This work is of high significance both scientifically and clinically as AURKA is a targetable kinase. Therefore, this work pushes the field forward and establishes a new molecule and pathway for investigation and intervention. They do this by using two mouse models of PKD: Inpp5e KO, a model of Joubert Syndrome (JS), and PJD KO, a model of adult onset PKD along with treatment with AURKA and AKT inhibitors. In both models AURKA deletion ameliorates the symptoms of PKD in these mice, cyst development, and cyst number and size. AURKA has a proven role in ciliogenesis and likewise deletion of AURKA reverses the effects of PKD on cilia number in the kidneys of these models. Further inhibition of AURKA and AKT phenocopies these results. They establish that AURKA has a significant role in cystogenesis and reduces the severity of adult onset PKD. Most importantly, they establish a novel reciprocal regulation by AKT. This reviewer recommends some changes in the Figures to better highlight their data and strengthen their case. Overall, this is an exceptionally well conducted study.

Figures

Figure 1. This figure would benefit from higher mag insets of the KO kidneys and double staining with DBA. The authors should discuss why the cystic index of the double KO is initially high and resolves at later time points. Is AURKA expressed at higher levels at early time points? A western would complement this data. Should define major and minor pathways in text.

Figure 2. Discuss the drop in rescue at P21 in d.

Figure 3. Discuss how the phenotype in mouse relates to that seen in human. Higher mag insets might be helpful for the reader and for interpretation. What happens for survival at longer time points? Including this is probably not necessary as the trend is clear but might be helpful.

Figure 4. Panel e graph, add single copy.

Figure 5. Panel p an q, show S473 results, for localization and intensity comparison.

Supplemental Figures

Figure S1. Double staining of AURKA and DBA should be shown along with higher mag insets. The KO of AURKA should be quantitated. A western would be more informative. The expression patterns in n should include wt as a control.

Figure S2. Include wt in addition to c, d, and e.

Figure S3. Include wt in a.

Figure S4. Where is panel L?

Reviewer #2 (Remarks to the Author):

In this detailed and comprehensive study, Tham et al describe a series of elegant genetic studies using two conditional knock-out mouse models of autosomal dominant polycystic kidney disease (PKD) and Joubert syndrome (JS), with floxed alleles of *Pkd1* and *Inpp5e*, respectively. Crosses with a *Hoxb7-Cre* driver line purported to lead to efficient conditional ablation in developing renal collecting duct lineages.

Crosses of the *Hoxb7-Cre* driver line with an Aurora kinase A (*Aurka*) floxed line gave a robust set of observations that *Aurka* is dispensable for collecting duct development. However, *Aurka* ablation causes a decrease in cyst formation and rescue of normal kidney function in both the conditional PKD and JS models. At the cellular level, *Aurka* loss causes rescue of normal cilia incidence, reduction of aberrantly high levels of proliferation and normalisation of DNA-damage responses in collecting duct epithelial cells in both the PKD and JS models. This data is convincing, consistent with the well-known function of AURKA as a mitotic kinase that interacts and/or phosphorylates proteins at the centrosomes and mitotic spindles (e.g. PLK1, TPX2, TACC3). The renal phenotypes are very well-characterised and are consistent with previously described *Inpp5e* models. Cysts form more easily when cell adherence is perturbed, and

it would be interesting to assess and document any adherence or cellular polarity changes in collecting duct epithelial cells (e.g. ZO-1 & moesin/ezrin expression & localization).

We are less convinced by the purported functional interaction with AKT. AURKA also has non-mitotic functions (ciliogenesis, regulation of N-MYC, DNA damage repair etc) and it is unclear if these depend solely on the kinase activity, or require interactions with intrinsic disordered regions (particularly in the N-terminus of the protein). AURKA interactors are often weak and transient, and this may be an explanation for the weak bands in Figure 4h & i. However, none of the AKT isoforms have previously been identified as substrates or interactants of AURKA (for example, the phosphoproteomics study by Kettenbach et al 2011, and Arslanhan et al 2021 used BioID proximity mapping to identify predominantly cell division, centriole satellite and centrosome proteins as potential AURKA interactants).

In terms of well-described AKT functions, on activation, AKT translocates from the cytosol to the plasma membrane where the PH domain binds to PIP3. Activation proceeds in stages, with phosphorylation of Thr308 in the kinase domain and then Ser473 in the C-terminal domain. The latter is mediated by many kinases, as well as by auto-phosphorylation. In Figure 5m, it is not clear if exogenous expression of AURKA leads to direct phosphorylation of Thr308 because the kinase-dead form (presumably dominant negative?) does not decrease phosphorylation levels. A cleaner model would be to use a cell-line in which AURKA is knocked-out or knocked-down rather than wild-type mIMCD3s. In parallel, a useful control would be to knock-down PTEN which should confirm that AKT can be over-activated in the model system.

In light of this previous data, we strongly recommend the authors to substantiate the AURKA-AKT interaction by an independent biochemical method. CoIPs using exogenously-expressed epitope-tagged proteins is the least burdensome approach, and a small series of deletion constructs should be used to refine the binding region in AURKA. (For example, there are implications to mechanism if the region is the IDR in the N-terminus vs the minimal interaction regions for TPX2 or TACC3). The authors should also substantiate that this interaction is necessary and sufficient for AURKA-dependent phosphorylation of AKT (e.g. using an in vitro kinase assay) because the current data does not rule out the indirect effect of other kinases or, indeed, AKT auto-phosphorylation.

In Figure 5-6, we would query the use of Alisertib/MLN8237 because, although it is widely used as an AURKA inhibitor (Ki 1.2nM), it does have off-target effects on AURKB. How is this potential confounding effect (due to inhibition of cell proliferation and cytokinesis) being mitigated and monitored? A selective small molecule inhibitor of AURKA such as LY3295668 (Ki 0.8 nM & 1038 nM for AURKA & AURKB, respectively) should be considered.

Minor points:

The introduction (para. 3) and discussion (para. 4) should provide some more background on AURKA biology, especially considering the purported direct functional interaction with the PI3K/AKT pathway. This is novel and quite unexpected, given the previous proteomic and proximity interactomes that have published (see above). Further background and context, in addition to further biochemical interaction studies, will help to make the case that this is a genuine pathomechanism across a range of ciliopathies.

In the cyst size dataset, *Inpp5e* Δ/Δ ; *Aurka* $\Delta/+$ double mutant are indistinguishable from *Inpp5e* Δ/Δ (Figure 1d). What is the “ns” value for the pairwise comparison so that it matches the results in Figure 1m.

We would question the efficiency of this conditional knock-out because Figure S1m (for a kidney cyst in the *Inpp5e* Δ/Δ ; *Aurka* Δ/Δ double mutant line) shows that there are still AURKA-expressing renal epithelial cells. Since there is still colocalization with DBA, this would imply that these are still collecting duct cells. Figure S1c does not reassure me that the Cre conditional has been that efficient, and the legend just states that “genomic DNA” has been analysed: presumably it is trivial to repeat this in DNA from kidney tissue?

Figure 4a: why is the comparison in the yellow set of the Venn diagram against *Pkd1*? This must be a typo, perhaps it is meant to be *Inpp5e*?

Figure 4h-i: why has the peptide block (lane 4) appeared to work well as a negative control in h) for *Pkd1* crosses but not in i)? In any case, these experiments need to be repeated with the inclusion of an independent biochemical approach to corroborate the potential AURKA-AKT interaction

Figure 5m-o: wandering or missing labels for the bar graphs

Suppl Dataset 2: in the enriched KEGG pathway datasets, the abbreviations "IKO", "AKO", "PAKO" and "IAKO" are not obvious and renaming them or a readme tab would be helpful to improve clarity

The shared KEGG pathways (Figure 4c) should reflect data that can be viewed in the suppl dataset 2. It is unclear how AKT signalling is identified as a shared dysregulated pathway from the data presented.

“Major and minor pathways” are a somewhat confusing nomenclature for different types of cystic kidney disease. (There are other genes for which mutations cause ADPKD other than polycystins, e.g.

GANAB, so how does these fit into this model?) This nomenclature does need some further explanation and comparison with the older idea of a switch between non-canonical/PCP vs canonical Wnt signalling during cytotogenesis. It would be useful to incorporate these ideas into a schematic of the proposed pathomechanisms and to present this in the end main figure: the schematic in Figure S7j is completely opaque without further explanation in the figure legend.

Reviewer #3 (Remarks to the Author):

This outstanding study by Tham et al. investigated the role of Aurora Kinase A (AURKA) in regulating renal cyst development. The authors examined the gene's role in several mouse models of Autosomal Dominant Polycystic Kidney disease (ADPKD) as well as of recessive Joubert Syndrome (caused by INPP5e) by conditional deletion of the gene at several time points. Whereas lack of AURKA alone did not alter the development of the renal collecting duct system, in both the PKD1 homozygous loss of function model as well as the INPP5e loss of function model, the renal cystic phenotype was mitigated when rendering those mice AURKA homozygously negative, in the adult ADPKD model even in a heterozygous state. The authors thereby convincingly demonstrate that loss of AURKA function protects from development of renal cysts in both models. In addition the authors examined in detail the role of AURKA in signaling pathways and demonstrate a converging role in the AKT pathway, as well as the contrary effect on cyst development by the AURKA kinase inhibitor Alisertib.

Major comments

There are no major concerns.

Minor comments

1. In figure 1j) please indicate the age of the dissected animals.
2. In figure 2 please also show control conditions of the IF stainings (control genotype, possibly images with secondary antibodies only).
3. In figure 4 a-c) please change coloring of inscriptions since it is barely readable.
4. In figure 4 e) please add labeling
5. Please explain explain abbreviations before using them in text (e.g. PC2 in line 82).

Reviewer #4 (Remarks to the Author):

In this manuscript Tham et al delineate the role of AURKA in mouse models of ADPKD and JS and suggest that its pathomechanistic role is mediated through AKT signaling. The authors show impressive, nearly complete inhibition of rapidly progressive PKD upon Aurka loss in the setting of Pkd1 or Inpp5e loss. This stunning disease inhibition was further confirmed in an adult ADPKD model. Based on RNAseq data the authors conclude that the mechanism of action is through AKT. They perform a series of experiments supporting this conclusion, including treatment with an AKT inhibitor. While the data shown on Aurka loss is truly striking and exciting, laying the foundation for the protein being a very attractive target in PKD therapeutics, the connection to AKT is less convincing. Multiple points in the data suggest that AKT might not be the mechanistic link between AURKA overexpression in PKD and the nearly complete inhibition of disease upon protein loss.

Major points of criticisms.

1. The authors build the paper on the hypothesis that the observed disease alleviation of CD Aurka loss in the setting of CD Inpp5e or Pkd1 loss is mediated by the common pathway AKT signaling or AURKA/AKT interaction. The data supporting this hypothesis is somewhat underwhelming.

a. It seems plausible that the pathomechanism underlying AURKA-mediated cystogenesis differs between Inpp5e and Pkd1. (a) Inpp5e CD KO animals continue to have cysts growth, although impressively reduced, post Aurka CD loss while Pkd1 CD KO animals had an astonishing nearly complete suppression of cystogenesis post Aurka CD loss. (b) As mentioned by the authors, the RNAseq analyses highlight a near correction of dysregulated genes upon CD Aurka loss on the Inpp5e background (although cyst growth remains), while CD Aurka loss on the Pkd1 background only mildly impacted the dysregulated gene networks associated with CD Pkd1 loss alone. Important to note is the previously well-established link between INPP5E and AURKA.

b. The IPs shown in Figure 4h and 4i are not very convincing. (a) The +V5 lane in "h" should have a much stronger detection level for V5 Ab than shown; similar as to "i"; (b) The band for AKT positivity is very weak and difficult to see. Further, the authors need to address the important question of why this interaction only occurs in the setting of PKD. What are possible underlying mechanisms?

c. The impact of AURKA in vitro siRNA inhibition/over-expression, or the use of kinase dead AURKA on pAKT/AKT are not very impressive in terms of magnitude. Similar, the Alisertib effects on AKT are mild (see point 6).

d. The PKD phenotype observed upon AKT inhibition is strikingly non-comparable to the one seen with CD Aurka loss. (i) AKT inhibition impacts both cyst size and number, while CD Aurka loss impacts only cyst number. (ii) Cystic disease is still very much present upon AKT inhibition, but nearly abolished with CD Aurka loss.

e. The authors report a reduction of AURKA upon AKT inhibition (Figure S7d) and hence suggest a transcriptional feedback regulation of AKT on AURKA. However, it is quite possible that the reduction of AURKA is caused by less severe disease rather than AKT inhibition.

f. To support the connection between AURKA-AKT better, it would be interesting to treat CD Aurka; Pkd1 KO and Aurka; Inpp5e KO mice with an AKT activator such as SC79.

2. The findings that CD Aurka KO nearly completely ameliorates PKD are astonishing and exciting. Based on the presented data that cyst number rather than cysts size is significantly impacted, it begs to hypothesize that AURKA is predominantly involved in cyst initiation versus growth. The authors have not addressed that nuance. It would be important to knock-out (or effectively pharmacologically inhibit, NOT with Alisertib) AURKA at a time point when cysts have already established in an adult model. Of note, the date shown in Figure 3 does not address this question, as both Pkd1 and Aurka are lost at the same time since the same cre driver was used. It is interesting to note though that in the adult model (Figure 3) heterozygous Aurka loss impacts cyst number in CD and distal tubules but not proximal tubules. The authors could speculate on this point in the discussion. In terms of the authors stating that the alleviation of disease happens in an AURKA-kinase independent way, more evidence (e.g. in vivo) would be required, as (a) the current data in the mIMCD3 cells is only moderately convincing and limited and (b) is done in WT cells albeit the authors suggest the connection between AURKA-AKT is only important in the setting of PKD.

3. It remains unclear and somewhat unaddressed why AURKA is upregulated in PKD (feedback between AURKA/AKT is mentioned very briefly but not convincingly, see point 1e). What is the driver? Is it transcriptional, translation, or dysregulation or proteins that regulate its expression? This is a key question for therapeutic translation, especially given the astonishing impact its deletion shows in the mouse models presented.

4. In respect to the AURKA overexpression and AURKA inhibitor studies: (a) The observation that Alisertib increases AURKA levels is surprising/interesting. This is in contradiction to earlier publications (Nikonova et al Front. Oncol. 2015). Can the authors please confirm this observation in the mouse studies? (b) The impact of Alisertib on pAKT is not convincing based on the provided WB. Further, the increase in pAKT does not seem significant enough to play a predominant role in driving worsening of PKD upon Alisertib administration (Figure 5h). This should also be confirmed in PKD cells, since the authors initially state that the AURKA/AKT interaction does only occur in the setting of PKD. (c) The data shown with the AURKA kinase dead construct are underwhelming. The changes stated are hard to see in the provided western blots and are very minimal. It is difficult to justify how this would drive significant biological changes. (d) Based on the provided data that both %KW/BW and cyst size is reduced by Alisertib, while cystic index remains unchanged, it is difficult to agree with the statement that Alisertib worsens PKD in the CD Inpp5e KO model, especially given the small N (3-5 animals) (i) It would be essential to test a different AURKA inhibitor, especially since multiple AURKA specific are FDA approved (MK-5108 has been tested in the setting of kidney fibrosis). (ii) given that the adult model seems more sensitive to AURKA levels, it would be important to test the drug in that model as an additional model.

5. It is plausible that inhibition of AKT is a therapeutic strategy for PKD. The authors show supporting evidence for this in two models. Since the N's for these studies are low and both models are rapidly progressing, it would require showing efficacy of AKT inhibition in a slowly progressive model as well. Although, this is somewhat tangential to this paper which in its predominance focuses on AURKA.

6. The authors show that CD Aurka loss in the setting of Inpp5e or Pkd1 loss corrects CD ciliation, DNA-damage responses and AKT-phosphorylation status. (a) In respect to ciliation and DNA-damage response, the authors fail to link how these may be mediated by AKT; AKT-ciliation; AKT- γ H2AX. (b) The authors also do not investigate whether these responses (ciliation, DNA damage response, AKT phosphorylation) are a direct effect of Aurka loss or an indirect effect of the kidney being no longer cystic. It would be important to redo the analyses in the remaining cysts within the double mutant kidneys.

7. Successful CD AURKA deletion needs to be confirmed in more detail e.g. western blot of flow sorted cells or IF staining at embryonic stages as expression should be higher at this time point. This is especially critical as the authors continue to see AURKA expressed in the CD cysts of PKD AURKA knock-out animals. Further, the authors need to settle on why this may be. They (a) justify it as AURKA positive “cysts derive from other cells (S1M)”, which would be surprising, since the cysts stains positive for DBA. INPP5E staining should be used to confirm this statement”; and (b) cysts “likely arise from incomplete cre activity”, which seems possible (S2d, Pkd1 model). Although, figure S2b only shows ~0.2% of CD cells express AURKA. It seems hard to believe that would be sufficient for driving cyst growth, given that for the case of CD Aurka/Pkd1 loss, the kidney looks otherwise indistinguishable from a wildtype.

8. The IF image of Figure 4d and 4e is a duplication of the exact same image. But the authors state mice of different genotypes were used.

Minor

1. The authors state that “AURKA expression was largely undetectable by P21” in control kidneys. This is somewhat surprising as other publications have shown it expressed by IHC or RNAseq in adult kidneys. Further, the authors themselves show no expression in Figure S1a, but show its expression in S1m. Both animals shown are WT P21.

2. The authors show that Alisertib treatment reduced ciliated cells. This is opposite to what has been published (Nikonova et al PNAS 2014). Could the authors comment?

3. When looking at the phosphorylation status of AKT in the setting of +/- CD Aurka loss (Figure 4d, e), it would be nice to differentiate between cystic and non-cystic tubules.

4. The authors show that cyst size in CD Pkd1/Aurka KO animals is >10 fold (0.5mm^2) compared to CD Pkd1 KO animals (0.05mm^2), Figure S2f; however, none of the histology sections shown, Figure 1j, n, show a single cyst in CD Pkd1/Aurka KO animals. It would be worth showing better representative images.

5. The images shown in Figure 6j don't seem to be representative images of the quantification provided in 6k.

6. The authors should double check the scale of Figure 7e. Cyst size reported in that figure is strikingly different from Figure 6 or Figure 1, which is the same model at the same age.

7. In multiple instances the figures or figure legends are missing important information or are inaccurate. (a) missing age of animals shown (histology/IF images) or analyzed (bar graph, time course

graphs, western blot); (b) stating incorrect labeling of stains performed; (c) describing panels that do not exist in the figure (e.g. S4L, should be j, k not k, l)

8. Figure S4d or the purpose for its analysis/depiction is not mentioned in the manuscript.

9. The indicated range of “N” for each experiment is very large in a variety of the figures (e.g. Figure 1f-h 3-13). It is very difficult to judge the robustness of the data based on this range. As required by multiple journals nowadays, I would suggest showing individual data points for each graph/quantification.

10. The authors make “quantitative statements” with regards to the IF images shown without providing further quantification results (e.g. increased cells expressing AURKA Figure S1m). The IF images shown are too high magnification to be able to critically evaluate if the statement made is true, even if quantification is provided. It is not sufficient to show an image of 7-12 cells and then claim something has quantitatively changed. Images obtained at smaller magnification need to be shown.

11. The loading in some western blots is off, hence the claimed results are difficult to judge (e.g. Figure 2j). It might be best to (a) show total protein (ponceau) or (b) representative images where loading is comparable. Also, it is unclear why the authors switch between baseline proteins throughout the paper (actin (S4g) vs. gapdh (S4b)).

12. It would be nice to see %KW/BW data for the animals analyzed in Figure 3.

13. I am not particularly keen on calling PKD1/PKD2 associated pathomechanisms driving PKD “major” pathway and ciliopathy gene associated pathomechanisms “minor” pathway. Per my impression this is not a common nomenclature among PKD scientists. Further, the pathways mediating disease are likely very complex and interconnected. Maybe the authors can consider changing this nomenclature. I presume they simply like to differentiate “dominant” and “recessive” disease or “non-syndromic” and “syndromic” PKD, but it is a bit unclear.

REVIEWER COMMENTS

Reviewer #1 (Remarks to the Author):

Introduction

In this manuscript, the authors seek to establish a link between Aurora A Kinase (AURKA), AKT, and Polycystic Kidney Disease (PKD). They present a highly convincing case that activity of this kinase is instrumental in the development of this devastating disease and its action is regulated by AKT not mTOR. This work is of high significance both scientifically and clinically as AURKA is a targetable kinase. Therefore, this work pushes the field forward and establishes a new molecule and pathway for investigation and intervention. They do this by using two mouse models of PKD: *Inpp5e* KO, a model of Joubert Syndrome (JS), and PJD KO, a model of adult onset PKD along with treatment with AURKA and AKT inhibitors. In both models AURKA deletion ameliorates the symptoms of PKD in these mice, cyst development, and cyst number and size. AURKA has a proven role in ciliogenesis and likewise deletion of AURKA reverses the effects of PKD on cilia number in the kidneys of these models. Further inhibition of AURKA and AKT phenocopies these results. They establish that AURKA has a significant role in cystogenesis and reduces the severity of adult onset PKD. Most importantly, they establish a novel reciprocal regulation by AKT. This reviewer recommends some changes in the Figures to better highlight their data and strengthen their case.

Overall, this is an exceptionally well conducted study.

Figures

Figure 1. This figure would benefit from higher mag insets of the KO kidneys and double staining with DBA.

These have been added to Figure 1a.

The authors should discuss why the cystic index of the double KO is initially high and resolves at later time points.

P6 added: "Over time the number of cysts reduced slightly as did the cystic index, possibly as a result of eventual Cre mediated Aurka deletion, a question we formally examine later in this article."

Using the *Pkd1* RC model (data Figure S3f-h), we now demonstrate AURKA deletion from extant cysts halts cyst growth.

Is AURKA expressed at higher levels at early time points? A western would complement this data.

AURKA is expressed at higher levels at P11 than P180. WB data is now provided as Supp Fig 7i.

Should define major and minor pathways in text.

The introduction P3 now reads "Increasing evidence suggests that two different pathways are involved in the development of PKD, referred to as the "major" and "minor" cystogenic pathways (Hwang et al 2019)". Their respective definitions then follow as before. We acknowledge that this nomenclature is not universally employed and for this reason we have clarified it's use throughout the revised manuscript.

Figure 2. Discuss the drop in rescue at P21 in d.

P8 added: "It is likely that this reduction is due to loss of INPP5E and its broader role in maintaining cilia stability independently of AURKA's actions."

Figure 3. Discuss how the phenotype in mouse relates to that seen in human.

P9 now reads "after this shift in adult mice (more analogous to ADPKD in humans),"

Higher mag insets might be helpful for the reader and for interpretation.

High magnification images for kidneys at 20 weeks of age have been added into Fig 3. Similar images at 18 weeks that were originally in Supp Fig 3a have been removed to avoid repetition.

What happens for survival at longer time points? Including this is probably not necessary as the trend is clear but might be helpful.

For humane reasons the study was capped at 20 weeks as the risk of spontaneous animal death due to cyst formation reached tolerable limits at this point. P9 now reads "**While mice lacking *Pkd1* rapidly succumbed to disease by the ethically allowed limit of the experiment (20 weeks of age), we found** that both hetero- and homozygous deletion of *Aurka* significantly improved the survival of mice with conditional mutations in *Pkd1* in a dose dependent manner"

Figure 4. Panel e graph, add single copy.

We have now included *Aurka* single copy (*Pkd1* Δ/Δ ; *Aurka* $\Delta/+$) data. The results show the expected intermediate level of % pAKT T308+ve CD cells between that of *Pkd1* Δ/Δ ; *Aurka* Δ/Δ and *Pkd1* Δ/Δ samples at P4.

Figure 5. Panel p an q, show S473 results, for localization and intensity comparison.

We have performed this analysis and added the data as Fig 5r. In short, pAKT S473 detection was largely absent from the basal body under these starved conditions, despite occasional detection in nuclei or in cells undergoing mitosis. Its profile is distinct to that of AKT pT308.

Supplemental Figures

Figure S1. Double staining of AURKA and DBA should be shown along with higher mag insets.

Higher magnification Insets have been added to Figure S1.

The KO of AURKA should be quantitated. A western would be more informative.

Tissue western blotting for endogenous murine AURKA, were not possible due to proximity of IgG bands generated from mouse-on-mouse cross-reactivity (combined with the limitations of this antibody and others trialled to date).

However, we would note, counts the % of AURKA+ve collecting duct cells in mice across a range of genotypes (Figures S1n and S2e at P21 and P4 respectively) shows a reduction in AURKA detection in both instances. Controls ~1.2-1.8% v *Aurka* KO 0.3% at P21, and Controls ~1.1-1.8% v *Aurka* KO ~0.5% at P4 puts the cre deletion efficiency in a range of 55%-83% which is wholly consistent with the originally reported 60-80% efficiency of this driver.

The expression patterns in n should include wt as a control.

This figure documents the expression patterns of cysts. As wild type mice do not have cysts there is no expression pattern to include.

Figure S2. Include wt in addition to c, d, and e.

A wild type image has been included (now (a)) however as noted above, e (now (f)) relates to the expression pattern of cysts, which are not present in wild type animals.

Figure S3. Include wt in a.

There isn't a wild type in this study, however we have included negative control (phenotypically normal) $Pkd1^{fl/fl}; Aurka^{fl/fl}; Pax8\text{-rtTA}$ animal which lacked Cre and did not develop ADPKD. Note this is now part of Figure 3a, as per our earlier response.

Figure S4. Where is panel L?

Apologies, there was no L in first submission. However, the revisions have now created a panel S4L.

Reviewer #2 (Remarks to the Author):

In this detailed and comprehensive study, Tham et al describe a series of elegant genetic studies using two conditional knock-out mouse models of autosomal dominant polycystic kidney disease (PKD) and Joubert syndrome (JS), with floxed alleles of *Pkd1* and *Inpp5e*, respectively. Crosses with a *Hoxb7-Cre* driver line purported to lead to efficient conditional ablation in developing renal collecting duct lineages.

Crosses of the *Hoxb7-Cre* driver line with an Aurora kinase A (*Aurka*) floxed line gave a robust set of observations that *Aurka* is dispensable for collecting duct development. However, *Aurka* ablation causes a decrease in cyst formation and rescue of normal kidney function in both the conditional PKD and JS models. At the cellular level, *Aurka* loss causes rescue of normal cilia incidence, reduction of aberrantly high levels of proliferation and normalisation of DNA-damage responses in collecting duct epithelial cells in both the PKD and JS models. This data is convincing, consistent with the well-known function of AURKA as a mitotic kinase that interacts and/or phosphorylates proteins at the centrosomes and mitotic spindles (e.g. PLK1, TPX2, TACC3). The renal phenotypes are very well-characterised and are consistent with previously described *Inpp5e* models.

Cysts form more easily when cell adherence is perturbed, and it would be interesting to assess and document any adherence or cellular polarity changes in collecting duct epithelial cells (e.g. ZO-1 & moesin/ezrin expression & localization).

We have assessed adherence/polarity with KSP and DBA localisation. KSP localised to the basal surface whilst DBA was enriched on the luminal surface, however this was unchanged in cysts from both *Inpp5e* and *Pkd1* neonatal KO models. On this basis we find no evidence for obvious defects in cell polarity. This data is presented in Figure S2h.

We are less convinced by the purported functional interaction with AKT. AURKA also has non-mitotic functions (ciliogenesis, regulation of N-MYC, DNA damage repair etc) and it is unclear if these depend solely on the kinase activity, or require interactions with intrinsic disordered regions (particularly in the N-terminus of the protein). AURKA interactors are often weak and transient, and this may be an explanation for the weak bands in Figure 4h & i. However, none of the AKT isoforms have previously been identified as substrates or interactants of AURKA (for example, the phosphoproteomics study by Kettenbach et al 2011, and Arslanhan et al 2021 used BioID proximity mapping to identify predominantly cell division, centriole satellite and centrosome proteins as potential AURKA interactants).

The predominant isoforms of AURKA detected upon V5-pulldown from cystic kidney extracts were smaller (N-terminally truncated). On this basis we do not believe the N-terminus of AURKA mediates interactions with AKT. To further explore this (and the interaction generally) we have now included an additional example of this interaction using testis extracts, where AURKA expression levels are considerably higher than in the kidney (see new Figure 4j), finding a strong interaction. While we readily acknowledge the studies noted by the reviewer, these studies were undertaken in immortalised cell lines (Hela and HEK293 cells respectively), and not from tissues - as in our work. Neither are renal epithelial cells (HEKs are likely neuronal). We also note that an interaction between AURKA and AKT has been previously detailed in FRET and Y2H screens (Li et al 2017 Pubmed ID:28205554, Arroyo et al 2014 Pubmed ID:24412244).

In terms of well-described AKT functions, on activation, AKT translocates from the cytosol to the plasma membrane where the PH domain binds to PIP3. Activation proceeds in stages,

with phosphorylation of Thr308 in the kinase domain and then Ser473 in the C-terminal domain. The latter is mediated by many kinases, as well as by auto-phosphorylation. In Figure 5m, it is not clear if exogenous expression of AURKA leads to direct phosphorylation of Thr308 because the kinase-dead form (presumably dominant negative?) does not decrease phosphorylation levels. A cleaner model would be to use a cell-line in which AURKA is knocked-out or knocked-down rather than wild-type mIMCD3s. In parallel, a useful control would be to knock-down PTEN which should confirm that AKT can be over-activated in the model system.

The kinase dead (K162R) form of AURKA doesn't act as a dominant negative but rather a variant that remains predominantly in the inactivated state (the K162R mutation likely interferes with the ATP pocket and T288 phosphorylation does not occur). Fig 5m demonstrates, over-expression of the inactive form is equally able to induce AKT pT308 phosphorylation compared to wild type over-expression. To clarify, we weren't attempting to show a decrease with knockdown but rather to highlight the 20% increase in pAKT levels which matches AURKA (WT) overexpression. This suggests the inactive AURKA can act as a scaffold for a complex that supports AKT activation. Given the observations of PDK1 localisation detailed in Fig 5q we would propose that this involves PDK1, but this has not formally been demonstrated.

With regard to cleaner models, AURKA knock-out cell lines are reported to die due to p53 apoptotic activation. We attempted to generate AURKA null IMCD3 cells using CRISPR, but observed apoptosis in cultures and a failure to generate engineered clones. Data from siRNA knock-down (at 50% which doesn't lead to cell death), is already included in Fig 5, however simultaneous knockdown and repletion with AURKA KD would still have some level of endogenous active AURKA present. For these reasons so we don't believe the results of such an experiment would be any clearer than the AURKA KD over-expression approached detailed in our original submission.

In light of this previous data, we strongly recommend the authors to substantiate the AURKA-AKT interaction by an independent biochemical method. CoIPs using exogenously-expressed epitope-tagged proteins is the least burdensome approach, and a small series of deletion constructs should be used to refine the binding region in AURKA. (For example, there are implications to mechanism if the region is the IDR in the N-terminus vs the minimal interaction regions for TPX2 or TACC3).

As detailed above we have now included unequivocal interaction data from testis extracts where AURKA is more strongly expressed, and we draw the reviewer's attention to other studies reporting such an interaction. Testis expresses full length AURKA and cystic kidneys express N-terminally truncated protein, yet both co-IP AKT. On this basis we conclude the N-terminus is not the contact site for the interaction and include this comment in the revised manuscript (Page 12).

The authors should also substantiate that this interaction is necessary and sufficient for AURKA-dependent phosphorylation of AKT (e.g. using an in vitro kinase assay) because the current data does not rule out the indirect effect of other kinases or, indeed, AKT auto-phosphorylation.

We believe PDK1 is mediating AKT T308 phosphorylation independent of AURKA's kinase activity. We are unaware of a commercial source for an AURKA kinase dead recombinant protein and performing classic in vitro kinase assays with available AURKA (WT) reagents wouldn't be informative, as keeping AURKA under kinase inactive conditions would equally prevent PDK1 kinase activity and AKT auto-phosphorylation.

In Figure 5-6, we would query the use of Alisertib/MLN8237 because, although it is widely used as an AURKA inhibitor (Ki 1.2nM), it does have off-target effects on AURKB. How is this potential confounding effect (due to inhibition of cell proliferation and cytokinesis) being mitigated and monitored? A selective small molecule inhibitor of AURKA such as LY3295668 (Ki 0.8 nM & 1038 nM for AURKA & AURKB, respectively) should be considered.

We have included Inpp5e + AURKA KO and AURKA KO animals in the Alisertib analysis and note that without AURKA, alisertib does not have pro-cystogenic effects. This confirms Alisertib acts via AURKA and not via AURKB or another target and we apologise for not making this distinction clearer in the manuscript. To clarify this point, we have now altered the text on P14 as follows:

In contrast, Inpp5e Δ/Δ ;Aurka Δ/Δ Alisertib-treated mice displayed no change in kidney to body weight ratio, cyst index, cyst number or cyst size (Fig 6B-E), confirming Alisertib requires AURKA (and not AURKB or other targets) to elicit these pro-cystogenic effects.

Minor points:

The introduction (para. 3) and discussion (para. 4) should provide some more background on AURKA biology, especially considering the purported direct functional interaction with the PI3K/AKT pathway. This is novel and quite unexpected, given the previous proteomic and proximity interactomes that have published (see above). Further background and context, in addition to further biochemical interaction studies, will help to make the case that this is a genuine pathomechanism across a range of ciliopathies.

We apologise for not having done so but were limited by character count. We have now made some additions to expand on these points.

Introduction (para. 3) has added: "In breast cancers, AURKA has been shown to cause resistance to PI3K/AKT/mTOR inhibitors by reactivating AKT through phosphorylation at Serine 473 (Donnella et al 2018).

Discussion paragraphs 4 and 5 have also been redrafted to better highlight the pertinent AURKA-AKT biology, which was previously (and we acknowledge awkwardly) split between paragraphs.

In the cyst size dataset, Inpp5e Δ/Δ ;Aurka $\Delta/+$ double mutant are indistinguishable from Inpp5e Δ/Δ (Figure 1d). What is the "ns" value for the pairwise comparison so that it matches the results in Figure 1m.

We apologise for this lack of consistency. The pair equivalent to that indicated as not significant "ns" in 1M, is now also "ns" for 1D (being $p < 0.13$). We have annotated 1D to reflect this.

We would question the efficiency of this conditional knock-out because Figure S1m (for a kidney cyst in the Inpp5e Δ/Δ ;Aurka Δ/Δ double mutant line) shows that there are still AURKA-expressing renal epithelial cells. Since there is still colocalization with DBA, this would imply that these are still collecting duct cells. Figure S1c does not reassure me that the Cre conditional has been that efficient, and the legend just states that "genomic DNA" has been analysed: presumably it is trivial to repeat this in DNA from kidney tissue?

Please refer to our response to a similar question from reviewer 1. We estimate the recombination efficiency is 55-83% based on IF counts in Fig S1n and S2e. This is consistent with the efficiency of HoxB7-Cre reported by other groups.

The PCR is derived from ear notch DNA as HoxB7-Cre is also active in the melanocytes of skin. It was not intended to be quantitative for recombination efficiency, only to make the point that the recombined allele can be detected in the presence of HoxB7-Cre.

Figure 4a: why is the comparison in the yellow set of the Venn diagram against Pkd1? This must be a typo, perhaps it is meant to be Inp5?

We are grateful to the reviewer for pointing this out – it is indeed a typographical error and has now been amended.

Figure 4h-i: why has the peptide block (lane 4) appeared to work well as a negative control in h) for Pkd1 crosses but not in i)?

The V5 block is a control which determines whether the V5-tag itself is responsible for the AKT Co-IP. The V5 blocking peptide did not affect this interaction (as in h) but also spontaneously aggregated into higher order complexes (as in i). Regardless of whether it was present as a small peptide or large complex, it still demonstrated the capacity to competitively block AURKA-V5 binding to the column. This confirms that AKT interactions occur via AURKA and not the V5-tag in both models.

In any case, these experiments need to be repeated with the inclusion of an independent biochemical approach to corroborate the potential AURKA-AKT interaction

We have now included interaction data from testis extracts as a site of higher AURKA expression (with corresponding increases in the co-IP of AKT) and we have flagged other reports of this interaction detected by yeast two hybrid and FRET.

Figure 5m-o: wandering or missing labels for the bar graphs

We apologise for this – these errors have now been corrected.

Suppl Dataset 2: in the enriched KEGG pathway datasets, the abbreviations "IKO", "AKO", "PAKO" and "IAKO" are not obvious and renaming them or a readme tab would be helpful to improve clarity

Thank you for the suggestion. We have added Read Me tabs to both supplementary data sets.

The shared KEGG pathways (Figure 4c) should reflect data that can be viewed in the suppl dataset 2. It is unclear how AKT signalling is identified as a shared dysregulated pathway from the data presented.

We have added a comparison tab to highlight shared pathway dysregulation.

“Major and minor pathways” are a somewhat confusing nomenclature for different types of cystic kidney disease. (There are other genes for which mutations cause ADPKD other than polycystins, e.g. GANAB, so how does these fit into this model?) This nomenclature does need some further explanation and comparison with the older idea of a switch between non-canonical/PCP vs canonical Wnt signalling during cytotogenesis. It would be useful to incorporate these ideas into a schematic of the proposed pathomechanisms and to present this in the end main figure: the schematic in Figure S7j is completely opaque without further explanation in the figure legend.

The “minor” and “major” pathway nomenclature has been coined in the field previously: <https://doi.org/10.1016/j.cub.2019.01.047> but we acknowledge is not universally employed.

In short, the major pathway refers to a cilia-dependent cystogenic pathway that activates in the absence of PC1 and PC2 and can be rendered less severe by removal of the cilia (such

as with compound deletion of KIF3a). The minor pathway refers to a second cystogenic driver that involves ciliopathy associated proteins, such as INPP5E and ARL13B. Trafficking proteins like TULP3 appear to contribute to both minor and major pathways. To date there has not been any literature on where GANAB might sit in this framework. We position AURKA and AKT signalling as a common convergence point of the major and minor cystogenic pathways and are responsible for cystogenesis. At that position AKT and AURKA likely regulate the balance of canonical and non-canonical Wnt/PCP signalling. However as this is speculative, we feel it would be premature to overlay Wnt signalling in our model.

We have however moved the model in Fig S7j to the main figures as Fig 7s and improved the figure legend description as suggested. We have also modified references to these pathways throughout the text to reflect their polycystin- or ciliopathy-associated mechanisms. If the reviewer felt strongly about it we would happily remove any reference to this nomenclature, but feel that the revised manuscript captures these differences.

Reviewer #3 (Remarks to the Author):

This outstanding study by Tham et al. investigated the role of Aurora Kinase A (AURKA) in regulating renal cyst development. The authors examined the gene's role in several mouse models of Autosomal Dominant Polycystic Kidney disease (ADPKD) as well as of recessive Joubert Syndrome (caused by INPP5e) by conditional deletion of the gene at several time points. Whereas lack of AURKA alone did not alter the development of the renal collecting duct system, in both the PKD1 homozygous loss of function model as well as the INPP5e loss of function model, the renal cystic phenotype was mitigated when rendering those mice AURKA homozygously negative, in the adult ADPKD model even in a heterozygous state. The authors thereby convincingly demonstrate that loss of AURKA function protects from development of renal cysts in both models. In addition the authors examined in detail the role of AURKA in signaling pathways and demonstrate a converging role in the AKT pathway, as well as the contrary effect on cyst development by the AURKA kinase inhibitor Alisertib.

Major comments

There are no major concerns.

Minor comments

1. In figure 1j) please indicate the age of the dissected animals.

We have added this information in a corrected in figure legend.

2. In figure 2 please also show control conditions of the IF stainings (control genotype, possibly images with secondary antibodies only).

We have now included the control cre only genotype in Figure S2, with staining for cilia, Ki67 and γ -H2AX for comparison (fig S2 i-k). They are included at reduced magnification to demonstrate the staining pattern(s) are not general.

We have also included secondary control images (without primary antibody but including DAPI, DBA-488 and anti-rabbit IgG-555 or anti-mouse IgG-555 taken from P11 cystic animals (Fig S2 l)). The lack of non-specific staining for anti-rabbit IgG-555 highlights the specificity of rabbit antibodies including pAKT T308, Ki67, Pericentrin and p53. As expected, the use of anti-mouse IgG-555 on mouse tissues does result in some background staining, most notably

in interstitial cells (presumably resident immune cells), however this did not generally occur in collecting duct cell epithelia, confirming the specificity of AURKA, acetylated-Tubulin and γ -H2AX profiles within the collecting ducts.

3. In figure 4 a-c) please change coloring of inscriptions since it is barely readable.

We have darkened the label colours to increase visibility.

4. In figure 4 e) please add labelling

This figure and associated labelling has been corrected in the revised manuscript.

5. Please explain abbreviations before using them in text (e.g. PC2 in line 82).

We have added POLYCYSTIN 2 (PC2) to the text in the revised manuscript.

Reviewer #4 (Remarks to the Author):

In this manuscript Tham et al delineate the role of AURKA in mouse models of ADPKD and JS and suggest that its pathomechanistic role is mediated through AKT signaling. The authors show impressive, nearly complete inhibition of rapidly progressive PKD upon Aurka loss in the setting of Pkd1 or Inpp5e loss. This stunning disease inhibition was further confirmed in an adult ADPKD model. Based on RNAseq data the authors conclude that the mechanism of action is through AKT. They perform a series of experiments supporting this conclusion, including treatment with an AKT inhibitor. While the data shown on Aurka loss is truly striking and exciting, laying the foundation for the protein being a very attractive target in PKD therapeutics, the connection to AKT is less convincing. Multiple points in the data suggest that AKT might not be the mechanistic link between AURKA overexpression in PKD and the nearly complete inhibition of disease upon protein loss.

We thank the reviewer for their comments. To be clear, we do not claim that AKT is the sole effector mechanism underpinning AURKA's role in cystogenesis, nor does our data support this view. Rather we contend that AKT is a contributing pathway. We view PKD as a complex disease mediated by several different regulatory pathways. While this explains the relatively modest rescue affected by AKT inhibition, it draws attention to the striking disease rescue realised by AURKA deletion – suggesting that this protein sits in a pivotal position as a regulator of disease.

Major points of criticisms.

1. The authors build the paper on the hypothesis that the observed disease alleviation of CD Aurka loss in the setting of CD Inpp5e or Pkd1 loss is mediated by the common pathway AKT signaling or AURKA/AKT interaction. The data supporting this hypothesis is somewhat underwhelming.

We apologise for the lack of clarity here. All that we are contending is that AURKA/AKT interactions are a part of the cystogenic pathway regulated by AURKA but we are not claiming that they are the only one. To acknowledge this, we have now added “functioning in-part via AKT” to the abstract.

a. It seems plausible that the pathomechanism underlying AURKA-mediated cystogenesis differs between Inpp5e and Pkd1. (a) Inpp5e CD KO animals continue to have cysts growth, although impressively reduced, post Aurka CD loss while Pkd1 CD KO animals had an astonishing nearly complete suppression of cystogenesis post Aurka CD loss. (b) As mentioned by the authors, the RNAseq analyses highlight a near correction of dysregulated genes upon CD Aurka loss on the Inpp5e background (although cyst growth remains), while CD Aurka loss on the Pkd1 background only mildly impacted the dysregulated gene networks associated with CD Pkd1 loss alone. Important to note is the previously well-established link between INPP5E and AURKA.

We agree that it is possible that mechanisms are different, but there is no question of AURKA's central role and of the role of AKT in both cases. Direct comparison of the different models in terms of phenotypic differences (as above) should be done with some care, given that the efficiency of recombination may differ at different genomic loci (Pkd1 vs Inpp5e) and because the temporal profile of recombination might also vary.

b. The IPs shown in Figure 4h and 4i are not very convincing. (a) The +V5 lane in "h" should have a much stronger detection level for V5 Ab than shown; similar as to "i";

Please note +V5 is +V5 blocking peptide and not +V5 Antibody. The Figure has been better labelled to remove this confusion.

(b) The band for AKT positivity is very weak and difficult to see. Further, the authors need to address the important question of why this interaction only occurs in the setting of PKD. What are possible underlying mechanisms?

We have now included data for the AKT-AURKA interaction from testis tissue extracts as a site of higher AURKA expression and, as one would expect, the efficiency of co-IP increases. In the undiseased kidney there is very little expression of AURKA. It is only when cystic disease is triggered by a lesion at another site (Pkd1, Inpp5e) that we see upregulation of the gene. The mechanism by which this occurs (i.e. how is AURKA expression regulated) is an open one. We have previously shown that *Aurka* expression is regulated by activated AKT signalling, a widely recognised feature of PKD. This may explain, to an extent, AURKA expression but it is almost certainly not the only reason why the protein is upregulated.

c. The impact of AURKA in vitro siRNA inhibition/over-expression, or the use of kinase dead AURKA on pAKT/AKT are not very impressive in terms of magnitude. Similar, the Alisertib effects on AKT are mild (see point 6).

This is a very subjective comment. What would be "impressive"? And why is absolute magnitude important? It is certainly significant. AKT signalling is a very potent cellular regulator and even small changes have important consequences. Also, PKD is a progressive condition characterised by gradual advances in disease severity - it is not cancer. We contend that changes of this magnitude can indeed be functionally important, and our experiments with AKT inhibitors argue strongly for this being the case.

d. The PKD phenotype observed upon AKT inhibition is strikingly non-comparable to the one seen with CD Aurka loss. (i) AKT inhibition impacts both cyst size and number, while CD Aurka loss impacts only cyst number. (ii) Cystic disease is still very much present upon AKT inhibition, but nearly abolished with CD Aurka loss.

These are very different experiments and directly comparing them in this way is misleading. AURKA deletion occurs embryonically in HoxB7-Cre models while AKT inhibition starts at P4 when the kidneys are already cystic. The reason for this is largely practical - AKT inhibitors

could not be given earlier as handling newborn pups promotes in-nest cannibalism or may have resulted in gestational toxicity. We also acknowledge that AURKA's effects are "in part" mediated by AKT, so the input of additional pathways in the overall phenotype is also expected. In the revised manuscript we have shown that *Aurka* deletion in pre-existing cysts results in reductions in cyst size (see below) – an experiment much more akin to those involving AKT inhibition.

e. The authors report a reduction of AURKA upon AKT inhibition (Figure S7d) and hence suggest a transcriptional feedback regulation of AKT on AURKA. However, it is quite possible that the reduction of AURKA is caused by less severe disease rather than AKT inhibition.

This is certainly possible, however we have shown in a previous publication (Plotnikova et al 2015) that AKT regulates *Aurka* transcription and our observations here are consistent with that finding.

f. To support the connection between AURKA-AKT better, it would be interesting to treat CD *Aurka*; *Pkd1* KO and *Aurka*; *Inpp5e* KO mice with an AKT activator such as SC79.

This is an interesting suggestion but ultimately such an experiment would be hard to interpret. While SC79 promotes AKT activity, it also prevents AKT translocation to PH domain mediated sites. Consequently, it would likely prevent co-localisation with AURKA at the basal body of primary cilia and poorly reflect the associated AKT-mediated signalling events that we believe transmit from that organelle. There would likely also be confounding implications of systemic activity to mouse health.

2. The findings that CD *Aurka* KO nearly completely ameliorates PKD are astonishing and exciting. Based on the presented data that cyst number rather than cysts size is significantly impacted, it begs to hypothesize that AURKA is predominantly involved in cyst initiation versus growth. The authors have not addressed that nuance. It would be important to knock-out (or effectively pharmacologically inhibit, NOT with Alisertib) AURKA at a time point when cysts have already established in an adult model. Of note, the date shown in Figure 3 does not address this question, as both *Pkd1* and *Aurka* are lost at the same time since the same cre driver was used.

In response to the reviewers very helpful suggestion we have performed exactly this study. Making use of the *Pkd1* "RC" mouse model - which has mild ADPKD driven by a hypomorphic allele identified in a human pedigree - and we now include the data on page 9, Fig S3f-h.

Using the RC background, we deleted *Aurka* using the inducible adult Dox system at 4 weeks of age and aged the mice to 6 months. We found that *Aurka* deletion completely prevented cyst growth (i.e. cysts were indistinguishable in size 6 months after deletion as compared with those at 4 weeks of age). While we did not note significant decreases in cyst number (within the constraints of the significant variation in this measure, in this model) in our hands adult RC mice principally model cyst growth as opposed to cystogenesis. We only note small and variable increases in cyst number between 4 weeks and 6 months of age.

Taken together with the other models detailed in the manuscript, our results show that AURKA is important both for cyst initiation and for cyst growth. Discussion of these results has been added on P17. Although it took a considerable period of time to generate these results, we think they add a critical aspect to our understanding of the role of AURKA in cyst biology and we thank the reviewer for suggesting the experiment.

It is interesting to note though that in the adult model (Figure 3) heterozygous Aurka loss impacts cyst number in CD and distal tubules but not proximal tubules. The authors could speculate on this point in the discussion.

This is an interesting question but not one for which we have a definitive answer. Perhaps the most likely explanation is that the different tubule cell types have different levels of AURKA expression, hence the requirement for a full complement of expression in cyst formation is different. We have therefore changed the text on P9 and added: "...It was noted this phenomenon was less evident in proximal tubules, which may relate to differences in AURKA expression between tubule types (Fig. S3c-e). ...".

In terms of the authors stating that the alleviation of disease happens in an AURKA-kinase independent way, more evidence (e.g. *in vivo*) would be required, as (a) the current data in the mIMCD3 cells is only moderately convincing and limited and (b) is done in WT cells albeit the authors suggest the connection between AURKA-AKT is only important in the setting of PKD.

We provide *in vivo* data on alisertib treated mice demonstrating the treatment causes an increase in both AURKA +ve and AKT pT308 +ve collecting duct cells in cystic animals (Fig 6K). Given, alisertib is the leading kinase inhibitor, it implies this *in vivo* effect is kinase independent, confirming similar (but more direct) *in vitro* observations with mIMCD3s.

mIMCD3 are an SV40 T antigen immortalised line and represent hyperproliferative collecting duct cells. In that regard, they are somewhat functionally akin to PKD cells in showing defects in polarity under spheroid formation assays.

3. It remains unclear and somewhat unaddressed why AURKA is upregulated in PKD (feedback between AURKA/AKT is mentioned very briefly but not convincingly, see point 1e). What is the driver? Is it transcriptional, translation, or dysregulation or proteins that regulate its expression? This is a key question for therapeutic translation, especially given the astonishing impact its deletion shows in the mouse models presented.

We agree that this is an interesting and therapeutically relevant question, but it is simply beyond the scope of the existing project.

4. In respect to the AURKA overexpression and AURKA inhibitor studies: (a) The observation that Alisertib increases AURKA levels is surprising/interesting. This is in contradiction to earlier publications (Nikonova et al Front. Oncol. 2015).

Our Aurka-V5 knock-in reporter mouse strain demonstrates that AURKA exists predominantly as smaller N-terminally truncated isoforms in mouse kidney extracts and any amount of full length AURKA is very small in comparison. The BD monoclonal antibody cited by Nikonova, is raised to an N-terminal peptide that is missing from these smaller isoforms. For these reasons, we directly count collecting duct cell % from immunostaining or WB from mIMCD3 cells lacking endogenous IgG. AURKA stabilisation appears to be a common feature of kinase inhibitors and in the revised manuscript we observe similar increase in cells treated with MK5108, VX680 and C1368 (see below).

Can the authors please confirm this observation in the mouse studies?

This was assessed by directly counting the number of AURKA+ve cells in the kidney collecting duct. Taking this approach avoids signal dilution and the potential confounding impact of other cell types which would be present in protein preparations from whole kidney Western blots. Our analysis shows twice as many detectable cells upon Alisertib treatment.

(b) The impact of Alisertib on pAKT is not convincing based on the provided WB. Further, the increase in pAKT does not seem significant enough to play a predominant role in driving worsening of PKD upon Alisertib administration (Figure 5h). This should also be confirmed in PKD cells, since the authors initially state that the AURKA/AKT interaction does only occur in the setting of PKD.

There were two inputs into the pAKT (T308) ratio being increased - pAKT and reduced total AKT. The mIMCD3 WB band of pAKT (T308) is larger and darker (i.e. increased) upon Alisertib treatment AND reduced total AKT levels giving the ratio outlined. As this is a shifting baseline it is less obvious but confirmed by densitometry. These observations have already been confirmed in cyst cells *in vivo* via counts of positive cell proportions from immunostaining. With respect, we would argue that in comparison, most PKD cell lines are relatively poor models of disease.

(c) The data shown with the AUKRA kinase dead construct are underwhelming. The changes stated are hard to see in the provided western blots and are very minimal. It is difficult to justify how this would drive significant biological changes.

Refer to response for point 1c regarding subjectivity. With respect, we would suggest that chronic upregulation of AKT signalling by 20-30% is very likely to be deleterious.

(d) Based on the provided data that both %KW/BW and cyst size is reduced by Alisertib, while cystic index remains unchanged, it is difficult to agree with the statement that Alisertib worsens PKD in the CD Inpp5e KO model, especially given the small N (3-5 animals).

We have reworded to state cystogenesis is worsened. This is broadly consistent with the worsening of disease by Alisertib in Pkd1 mouse models.

(i) It would be essential to test a different AURKA inhibitor, especially since multiple AURKA specific are FDA approved (MK-5108 has been tested in the setting of kidney fibrosis).

We have now examined MK5108 under the same mIMCD3 48hr culture conditions and gone a step further in examining two additional AURKA kinase inhibitors, VX680 and C1368. With all these inhibitors we see AURKA upregulation and increases in pAKT (T308) as we observed with Alisertib. This is now included as Fig S5h-j. This suggests most ATP-competitive AURKA inhibitors including MK5108, VX680 and C1368 also share kinase-independent agonist properties like Alisertib. Testing these models in the current study, which now includes an additional assessment of AURKA function in existing cysts, is beyond the scope of this study. However, we agree in principle that it would be interesting to compare these different inhibitors *in vivo*.

(ii) given that the adult model seems more sensitive to AUKRA levels, it would be important to test the drug in that model as an additional model.

Alisertib-treatment of adult ADPKD models has already been described (Nikonova et al Front. Oncol. 2015) and (Nikonova et al PNAS 2014). The conclusions of those papers were also that Alisertib exacerbated ADPKD. Again, we agree this is an interesting question, but it is beyond the scope of the current work.

5. It is plausible that inhibition of AKT is a therapeutic strategy for PKD. The authors show supporting evidence for this in two models. Since the N's for these studies are low and both models are rapidly progressing, it would require showing efficacy of AKT inhibition in a slowly progressive model as well. Although, this is somewhat tangential to this paper which in its predominance focuses on AURKA.

As the reviewer has acknowledged, AKT inhibition in the adult model is outside of the remit of this study.

6. The authors show that CD Aurka loss in the setting of Inpp5e or Pkd1 loss corrects CD ciliation, DNA-damage responses and AKT-phosphorylation status. (a) In respect to ciliation and DNA-damage response, the authors fail to link how these may be mediated by AKT; AKT-ciliation; AKT- γ H2AX. (b) The authors also do not investigate whether these responses (ciliation, DNA damage response, AKT phosphorylation) are a direct effect of Aurka loss or an indirect effect of the kidney being no longer cystic. It would be important to redo the analyses in the remaining cysts within the double mutant kidneys.

AKT-mediated DNA damage responses are already known: [10.1093/hmg/ddz232](https://doi.org/10.1093/hmg/ddz232)

The mechanism of AKT-mediated ciliation represents a difficult question to directly address. Are the cells less proliferative resulting in an increase in ciliation or is cilia resorption reduced, resulting in less proliferation? Addressing this question is non-trivial. We acknowledge that better understanding the cellular events driving disease rescue is important, but it is simply beyond the scope of this work.

As we detail later in this response, the remaining cysts are likely a reflection of mosaic Cre activity (i.e. those cysts typically still express AURKA). By all measures they are identical to cysts from single KO animals. As they still exhibit AURKA staining, their analysis does not provide any insight as to whether AURKA loss or disease suppression regulates AKT as proposed.

7. Successful CD AURKA deletion needs to be confirmed in more detail e.g. western blot of flow sorted cells or IF staining at embryonic stages as expression should be higher at this time point. This is especially critical as the authors continue to see AURKA expressed in the CD cysts of PKD AURKA knock-out animals. Further, the authors need to settle on why this may be. They (a) justify it as AURKA positive “cysts derive from other cells (S1M)”, which would be surprising, since the cysts stains positive for DBA. INPP5E staining should be used to confirm this statement”; and (b) cysts “likely arise from incomplete cre activity”, which seems possible (S2d, Pkd1 model). Although, figure S2b only shows ~0.2% of CD cells express AURKA. It seems hard to believe that would be sufficient for driving cyst growth, given that for the case of CD Aurka/Pkd1 loss, the kidney looks otherwise indistinguishable from a wildtype.

These are important questions and we refer the reviewer to the comments we made in response to reviewer 1. Based on counts of immunostained cells we estimate the recombination efficiency is 55-83% (Fig S1n and S2e) which is consistent with the known efficiency of HoxB7-Cre. We have shown that remaining cysts express AURKA and we have shown they are similar to those in mice without AURKA deletion (added supplementary data S2m-r and S4a,b). The differences in degree of rescue most likely reflect that hierarchy of Cre efficiency. It is widely acknowledged that the activity of a given Cre varies depending on the locus being recombined. The simplest interpretation of our results is therefore that the Cre deletes Aurka better than Pkd1 (so most Pkd1 deleted cells are also Aurka deleted) whereas in the Inpp5e KO model, the Cre deletes Inpp5e better than Aurka (leaving a small proportion of Inpp5e KO only cells).

8. The IF image of Figure 4d and 4e is a duplication of the exact same image. But the authors state mice of different genotypes were used.

This is an embarrassing error for which we profoundly apologise (they are quite complex figures). The draft placeholder has now been updated/corrected to reflect the genotypes.

Minor

1. The authors state that “AURKA expression was largely undetectable by P21” in control kidneys. This is somewhat surprising as other publications have shown it expressed by IHC or RNAseq in adult kidneys. Further, the authors themselves show no expression in Figure S1a, but show its expression in S1m. Both animals shown are WT P21.

We agree with the reviewer that this is a somewhat misleading (and subjective) statement. We have rephrased this as “Whilst by P21 AURKA expression was limited to a small number of collecting duct cells...”. Note we have also restructured this figure to make space for high magnification images in Fig S1A/B as requested by another reviewer.

RNAseq and IHC staining also indicates zero to very low AURKA expression in human kidney according to the human proteinatlas.org.

2. The authors show that Alisertib treatment reduced ciliated cells. This is opposite to what has been published (Nikonova et al PNAS 2014). Could the authors comment?

These are two very different PKD models and making a direct comparison is difficult. It is entirely possible that the differences relate to 1) the genetic driver of disease, 2) Cre-target tissue and/or, perhaps most likely, 3) *in vitro* versus *in vivo* assessments of ciliation.

3. When looking at the phosphorylation status of AKT in the setting of +/- CD Aurka loss (Figure 4d, e), it would be nice to differentiate between cystic and non-cystic tubules.

This is a good suggestion. We have added a breakdown in terms of cystic and non-cystic tubules where possible (Figure S4a,b). Note however, there are no cystic tubules in WT, cre, double flox, and Aurka single KO animals, so these genotypes are not included. Also, at P4 there are not yet cystic tubules in *Inpp5e^{ΔΔ};Aurka^{ΔΔ}* mice.

We have also altered the text on Page 11 and added: “Furthermore, partitioning this data by cystic and non-cystic tubules demonstrated the elevation of pAKT (T308) was localised within cystic tubules (Fig S4a,b).”

4. The authors show that cyst size in CD *Pkd1/Aurka* KO animals is >10 fold (0.5mm^2) compared to CD *Pkd1* KO animals (0.05mm^2), Figure S2f; however, none of the histology sections shown, Figure 1j, n, show a single cyst in CD *Pkd1/Aurka* KO animals. It would be worth showing better representative images.

The reviewer appears to be mistakenly comparing cysts of *Pkd1/Aurka* KO animals at **150 days of age** with *Pkd1* KO animals at **11 days of age**. The data in Figure 1m instead shows a size difference of only **2 fold** ($\sim 0.05\text{mm}^2$ for *Pkd1* KO animals v 0.025mm^2 for *Pkd1/Aurka* KO animals). The relevant H&E images are given in 1J and are representative images. Such mice had 0-3 cysts. The histological sections that support S2f are present as Figure 1n.

5. The images shown in Figure 6j don't seem to be representative images of the quantification provided in 6k.

Figure 6K is a graph of % and not of cells per field. When you consider that a relatively low percentage of cells are positive for the marker(s) indicated, a single image will only show 0-2 positive cells when acquired at sufficient resolution. For this reason, the images presented are representative only. Graphs were generated across assessment of at least 10 random fields of view. This detail has been added to the methods.

6. The authors should double check the scale of Figure 7e. Cyst size reported in that figure is strikingly different from Figure 6 or Figure 1, which is the same model at the same age.

Figure 7e shows the same cyst size as Figure 1m, ranging from $\sim 0.05\text{mm}^2$ for Pkd1 KO animals compared with $\sim 0.04\text{mm}^2$ for Pkd1 KO animals treated with a cyclodextrin vehicle. The small difference would not be statistically significant and within system variability. Cyclodextrins are also known to have minor renal toxicity which could be an additional source of difference. Figure 6 is of an entirely different model, being the Inpp5e Kos, and is not comparable with Fig 1M or 7e.

7. In multiple instances the figures or figure legends are missing important information or are inaccurate. (a) missing age of animals shown (histology/IF images) or analyzed (bar graph, time course graphs, western blot); (b) stating incorrect labeling of stains performed; (c) describing panels that do not exist in the figure (e.g. S4L, should be j, k not k, l)

These instances have been corrected but note S4j,k have changed to S4l,m in revision.

8. Figure S4d or the purpose for its analysis/depiction is not mentioned in the manuscript.

Note S4d has become S4f in revision.

Page 11 now reads: "Upregulation of AKT expression, an increase in the AKT pT308/total AKT ratio and **downstream** pathway activity (via **p4EBP1 T37/46**) was further confirmed by AlphaLISA assays in the JS murine model and these parameters were normalised upon Aurka deletion (Fig S4d-f).

9. The indicated range of "N" for each experiment is very large in a variety of the figures (e.g. Figure 1f-h 3-13). It is very difficult to judge the robustness of the data based on this range. As required by multiple journals nowadays, I would suggest showing individual data points for each graph/quantification.

The n number has been split out per genotype and provided as a supplementary file.

10. The authors make "quantitative statements" with regards to the IF images shown without providing further quantification results (e.g. increased cells expressing AURKA Figure S1m).

This has been quantified and is provided as Fig S1n.

The IF images shown are too high magnification to be able to critically evaluate if the statement made is true, even if quantification is provided. It is not sufficient to show an image of 7-12 cells and then claim something has quantitatively changed. Images obtained at smaller magnification need to be shown.

The incidence of such cells is relatively low to the extent that providing images at sufficiently low magnification would be of little point. We have added this statement to the methods to make our approach to quantifying these measures clearer: “For in vivo cell proportions, at least 10 random fields across the kidney section were imaged and the number of positive cells for a given factor (Cilia, Ki67+ve, γ -H2AX +ve, AURKA+ve, AKT pT308 +ve, p53+ve etc) counted and divided by the number of collecting duct cells determined (across all images) using ImageJ.”

Furthermore, the point of the immunostaining is to show what positive and negative staining looks like at a cellular level - they were not intended to be taken as a singular image analysed to generate the accompanying graph.

11. The loading in some western blots is off, hence the claimed results are difficult to judge (e.g. Figure 2j). It might be best to (a) show total protein (ponceau) or (b) representative images where loading is comparable.

The actin loading of the lanes 3-6 are equivalent in this figure and include knock outs and control animals. They are therefore sufficient to demonstrate that only Inpp5e KO animals demonstrate γ -H2AX signals.

Also, it is unclear why the authors switch between baseline proteins throughout the paper (actin (S4g) vs. gapdh (S4b)).

As per the figure legend, S4d-f) were AlphaLISA assays, while S4g-i) were Western blots. The difference in baseline protein simply reflects the different assays. Given that all substantive comparisons are within (and not between) such assays, this makes no demonstrable difference to the outcome of our studies or their interpretation. Note that these figures are now S4i and S4d in revision.

12. It would be nice to see %KW/BW data for the animals analyzed in Figure 3.

This is a good suggestion and has now been included as Fig S3a.

13. I am not particularly keen on calling PKD1/PKD2 associated pathomechanisms driving PKD “major” pathway and ciliopathy gene associated pathomechanisms “minor” pathway. Per my impression this is not a common nomenclature among PKD scientists. Further, the pathways mediating disease are likely very complex and interconnected. Maybe the authors can consider changing this nomenclature. I presume they simply like to differentiate “dominant” and “recessive” disease or “non-syndromic” and “syndromic” PKD, but it is a bit unclear.

Please refer to response to reviewer 2 for the definition of major and minor pathways. We acknowledge this isn't uniformly adapted nomenclature and have made a number of changes throughout the document to clarify what we mean in discussing these pathways.

REVIEWER COMMENTS

Reviewer #1 (Remarks to the Author):

The authors have completely addressed my concerns in their revised manuscript.

Reviewer #2 (Remarks to the Author):

"Deletion of Aurora Kinase A prevents the development of Polycystic Kidney Disease"

In the revision of Tham et al., the authors have gone to considerable efforts to address my own comments, and those of the other three reviewers. This is an outstanding study that will have a significant impact on the PKD, ciliopathy and Aurora kinase fields. Their observation that conditional loss of AURKA causes amelioration of PKD is both interesting and exciting. It provides a provocative new set of therapeutic targets for PKD treatment.

I appreciated the inclusion of the following in this revision:

- 1) new data for endogenous coIPs to further substantiate the AURKA-AKT interaction (Figure 4j), as a second biochemical assay
- 2) data to confirm efficiency of conditional knockouts with the HoxB7-Cre driver
- 3) inclusion of new explanatory background on relevant AURKA-AKT biology in the Discussion
- 4) schematic Figure 7s and legend to clarify the polycystin- or ciliopathy-associated mechanisms in the "major" and "minor" disease pathways. I don't suppose that the actual labels matter, so long as the differences and nomenclatures are clear; it is undeniably useful to have this difference articulated in this paper
- 5) important new data that demonstrates AURKA is important both for cyst initiation and for cyst growth

Reviewer #3 (Remarks to the Author):

The authors thoroughly addressed all concerns and comments and convincingly strengthened their findings, therefore improving their study. There are no further concerns.

Reviewer #4 (Remarks to the Author):

Overall, the authors have significantly revised the manuscript and gathered new experimental data that significantly improves the quality and robustness of the study. However, some open concerns remain that would be beneficial to have addressed experimentally or by revising the manuscript text.

(1) The Aurka KO data in the adult models is interesting, especially considering the “dosage” effect. I appreciate the reviewers added the Pkd1 RC model as an additional model to establish the role of AURKA in adult disease, and a disease state at which PKD is “established”, akin to the AKT treatment approach. However, I find the presented data lack detail/explanation. (a) It would be important to show %KW/BW and cystic index. (b) It would also be informative to assess the various AKT-signaling associated parameters in this model. (c) It is interesting why in this model, compared to the Pax8-Pkd1 model, no effect on PKD severity is observed upon heterozygous loss of Aurka. Could the authors comment? (d) It is difficult to make firm conclusions that Aurka loss impacts PKD severity in the model based on the presented data. Cyst number is not altered, and cyst size only marginally changed in the control, and, as quantified seems higher prior Aurka loss (pre-dox, P28), compared to 26wks post Aurka loss (triangles), which would mean cyst shrunk in size post gene loss.

Please also explain in the method how Aurka loss was induced in this model.

(2) The authors make a strong and conclusive statement that only pAKT T308 is altered in disease and not pAKT S473, and that this is hence likely the key mechanistic link re Aurka function. This is supported in Figure S4d which shows no increase in %pAKTS473 in CD cells in Inppe5 mutants (12%) vs WT (11%), but Figure S8c (untreated, white bar; Inppe5 WT: 2%, KO 8% = 4-fold increase) does show a significant increase? Why is that discrepancy? I understand different ages were analyzed. But if pAKT S473 changes with more progressive disease P4 vs P15, then the authors should acknowledge that the mechanism of Aurka may go through phosphorylation at this site as well and additional timepoints would need to be analyzed in Figure S4. Please also quantify these parameters in the Pkd1 mutant treated with MK2206 (Figure S8).

(3) The authors state that polarity of epithelial cells is not impacted by Aurka loss, but the data shown in Figure S2h are very limited. Additional experimental approaches should be taken to evaluate polarity more thoroughly, at minimum by staining for Zo-1/Scrib and showing a z-stack of tubular regions.

(4) The presented BUN data of the adult-onset Pax8-Pkd1 model is confusing for the 20wk time point. The measured values between Figure 3e and S3b are quite different although both include animals at 20wks. Could the author's comment.

(5) It would be important to confirm that Alisertib increases AURKA levels in vivo. I requested this previously and the authors stated that it was "assessed by directly counting the number of AURKA+ve cells in the kidney collecting ducts" +/- treatment. However, I cannot find this data in the revised manuscript. It seems only AURKA + pAKT (T308) cells were quantified. Not AURKA alone.

(6) I feel the Figure title of Figure 6 is still an overstatement considering the low N of the data (N=3) and the very inconclusive results. KW/BW and cyst size decrease upon Alisertib treatment and there is only a modest increase in cyst number, nonetheless the title states the drug increases cystogenesis. Further, since the pro-cystogenic effect is not conclusive and Aurka KO mice don't have pronounced cystic disease, it is an overstatement to say that Alisertib acts through Aurka vs. AurkB.

(7) The manuscript would benefit from a discussion paragraph outlining the limitations of the study, explanations for discrepancies, and future directions. For example, (a) to emphasize better that AKT is only "in part" mediating the effect as shown by striking differences in KO vs treatment and highlight how the Aurka and AKT data cannot be directly compared (germline loss vs treatment at existing cysts). (b) PKD1 vs INpp5e differences in phenotype due to cre recombination efficiency. (c) Discussion on why AURKA is upregulated in PKD (d) Future direction of testing other AKT inhibitors in models or testing AKT inhibitor in adult model.

(8) I don't think it is appropriate that the exact animal number per bar graph are buried in a supplemental excel table. Use single data points in the graphs or writing the "n" under/above bar graph is more appropriate to better judge the data. Also, no numbers are provided for S2h-r in the supplemental excel table. Further, no gender info per n is given.

(9) S7 to be quantified, please

(10) Figure S8i is not referenced.

Reviewer #1 (Remarks to the Author):

The authors have completely addressed my concerns in their revised manuscript.

We thank reviewer 1 for their constructive input and positive recommendation.

Reviewer #2 (Remarks to the Author):

"Deletion of Aurora Kinase A prevents the development of Polycystic Kidney Disease" In the revision of Tham et al., the authors have gone to considerable efforts to address my own comments, and those of the other three reviewers. This is an outstanding study that will have a significant impact on the PKD, ciliopathy and Aurora kinase fields. Their observation that conditional loss of AURKA causes amelioration of PKD is both interesting and exciting. It provides a provocative new set of therapeutic targets for PKD treatment.

I appreciated the inclusion of the following in this revision:

- 1) new data for endogenous colPs to further substantiate the AURKA-AKT interaction (Figure 4j), as a second biochemical assay*
- 2) data to confirm efficiency of conditional knockouts with the HoxB7-Cre driver*
- 3) inclusion of new explanatory background on relevant AURKA-AKT biology in the Discussion*
- 4) schematic Figure 7s and legend to clarify the polycystin- or ciliopathy-associated mechanisms in the "major" and "minor" disease pathways. I don't suppose that the actual labels matter, so long as the differences and nomenclatures are clear; it is undeniably useful to have this difference articulated in this paper*

We have added major and minor labels to the figure 7s for further clarity.

- 5) important new data that demonstrates AURKA is important both for cyst initiation and for cyst growth*

We thank reviewer 2 for their constructive input and positive recommendation.

Reviewer #3 (Remarks to the Author):

The authors thoroughly addressed all concerns and comments and convincingly strengthened their findings, therefore improving their study. There are no further concerns.

We thank reviewer 3 for their constructive input and positive recommendation.

Reviewer #4 (Remarks to the Author):

- (1) The Aurka KO data in the adult models is interesting, especially considering the "dosage" effect. I appreciate the reviewers added the Pkd1 RC model as an additional model to establish the role of AURKA in adult disease, and a disease state at which PKD is "established", akin to the AKT treatment approach. However, I find the presented data lack detail/explanation.*

It would be important to show %KW/BW and cystic index.

Kidney to body weight ratios showed a significant reduction in *Aurka^{ΔΔ};Pkd1^{RC/RC}* animals relative to all other groups at 26 weeks. This data has now been included in a new supplementary figure 4 (splitting it out from the previous supplementary figure 3).

While cystic Index trended towards a reduction in *Aurka^{ΔΔ};Pkd1^{RC/RC}* animals relative to all other groups at 26 weeks, this only reached statistical significance for the No Dox v Dox comparison. This probably reflects the acknowledged variability in phenotypes in RC animals (see PMID: 33705824) and also the established nature of cystic disease in these mice at the point of *Aurka* deletion. Again, this data has been included in a new supplementary figure 4.

It would also be informative to assess the various AKT-signaling associated parameters in this model.

Our aim in examining this model was to establish whether *Aurka* deletion could arrest cyst progression and the data supporting this conclusion is very clear. We feel we have comprehensively addressed the roles of AURKA in regulating AKT signalling elsewhere in the manuscript. As it is, this analysis is challenging *in vivo* and the baseline variability in phenotypic presentation in RC mice would likely compound this.

It is interesting why in this model, compared to the Pax8-Pkd1 model, no effect on PKD severity is observed upon heterozygous loss of Aurka. Could the authors comment?

We agree that this is an interesting observation and speculate that it might reflect different cellular dependencies on AURKA with respect to cyst initiation relative to cyst progression. We have added this to the revised manuscript in the Discussion.

It is difficult to make firm conclusions that Aurka loss impacts PKD severity in the model based on the presented data. Cyst number is not altered, and cyst size only marginally changed in the control, and, as quantified seems higher prior Aurka loss (pre-dox, P28), compared to 26wks post Aurka loss (triangles), which would mean cyst shrunk in size post gene loss. Please also explain in the method how Aurka loss was induced in this model.

As we explain in our revision, the Pkd1 RC mice model the impact of *Aurka* deletion on extant cysts and we were not expecting there to be a significant reduction in cyst number (clarified on page 10, paragraph 1). This is because *Aurka* deletion is timed to occur after most cysts have formed. The apparent increase in cyst number over time when comparing P28 to 26 weeks likely reflects the sensitivity of detection - where a subset of developmentally induced cysts are too small to distinguish from normal tubules at P28 but grow to a detectable threshold by 26 weeks. This point has been added to the Discussion. There is no statistical difference in control cyst size between P28 and 26 weeks. The method for *Aurka* deletion in RC mice was also explained on Page 9, paragraph 2, however we have further clarified this in the Methods in the

revised manuscript. We also include Fig S4a to justify the time points selected for analysis and pooling of genders in the RC model.

(2) The authors make a strong and conclusive statement that only pAKT T308 is altered in disease and not pAKT S473, and that this is hence likely the key mechanistic link re Aurka function. This is supported in Figure S4d which shows no increase in %pAKTS473 in CD cells in Inppe5 mutants (12%) vs WT (11%), but Figure S8c (untreated, white bar; Inppe5 WT: 2%, KO 8% = 4-fold increase) does show a significant increase? Why is that discrepancy? I understand different ages were analyzed. But if pAKT S473 changes with more progressive disease P4 vs P15, then the authors should acknowledge that the mechanism of Aurka may go through phosphorylation at this site as well and additional timepoints would need to be analyzed in Figure S4. (Now Figure S5)

As outlined on Page 11, Paragraph 2, Line 5, both residues were altered in advanced disease, but we sought to investigate whether they were also early features. Only T308 changes were noted early in disease development. The point we were trying to make is that S473 likely marks late-stage disease but does not appear to be an early driver of pathogenesis. This is consistent with our previously published work (see PMID: 27056978 and PMID: 31625572). In the revised manuscript Fig S9c is a late stage time point and the comparable early time is illustrated in Fig 4d.

As late stage AKT activity has been previously published, the main premise of this paper is that AURKA and AKT pT308 mark a cystogenic switch; an early event in the transition from a normal tubule to a cyst. S473, in contrast, is not dysregulated at this point and only becomes so after cysts have advanced. We did not mean to suggest that AURKA has no secondary roles in regulating S473 but rather that this is not a feature of early disease. We have clarified this in the revised manuscript.

Please also quantify these parameters in the Pkd1 mutant treated with MK2206 (Figure S8) (now S9).

Figure 7g,h and S9 c,d, (Inpp5e model) establishes MK2206 acts on both AKT pT308, AURKA and pS473, however as we establish in this study, only the pT308 residue is regulated by AURKA in this context.

As such, the Pkd1 model analysis was focused only on quantifying pT308 and AURKA cells (Figure 7p) and we feel inclusion of pS473 data does not add anything to the story (nor was it requested in the first revision).

(3) The authors state that polarity of epithelial cells is not impacted by Aurka loss, but the data shown in Figure S2h are very limited. Additional experimental approaches should be taken to evaluate polarity more thoroughly, at minimum by staining for Zo-1/Scrib and showing a z-stack of tubular regions.

Previous studies by the Peters group have shown that ZO-1 is “abundantly expressed and correctly localised” in mouse models of ADPKD” (see PMID: 18666097). We have included an examination of changes in polarity (or lack thereof) at the request of Reviewer 2, but these studies are really outside of the remit of this work, and they do not impact substantively on our findings.

(4) The presented BUN data of the adult-onset Pax8-Pkd1 model is confusing for the 20wk time point. The measured values between Figure 3e and S3b are quite different although both include animals at 20wks. Could the author's comment.

We apologise for the lack of clarity here. As stated on page 9 (end Paragraph 1) and in Figure 3 (legend), the BUN data shown in Fig 3e is an end point BUN while that in Fig S3b is a time point BUN. As per figure legend, the end point is defined as either the conclusion of 20-week time course or at the point of humane euthanasia if earlier. This point has been added to the main text.

(5) It would be important to confirm that Alisertib increases AURKA levels in vivo. I requested this previously and the authors stated that it was "assessed by directly counting the number of AURKA+ve cells in the kidney collecting ducts" +/- treatment. However, I cannot find this data in the revised manuscript. It seems only AURKA + pAKT (T308) cells were quantified. Not AURKA alone.

The data to which the reviewer refers is similar to that of dual AURKA and AKT pT308 presented initially – so we didn't initially include it. We have now done so, both with respect to the individual %AURKA and %pAKT T308 data - in Supplementary Figure 7a and b. This data further confirms that the %AURKA +ve cells increase with Alisertib treatment and that a subset of these cells are also pAKT T308 positive. Conversely, almost all pAKT T308 positive cells are also AURKA positive.

(6) I feel the Figure title of Figure 6 is still an overstatement considering the low N of the data (N=3) and the very inconclusive results. KW/BW and cyst size decrease upon Alisertib treatment and there is only a modest increase in cyst number, nonetheless the title states the drug increases cystogenesis.

The results are statistically significant and concur with the findings of others that Alisertib worsens disease in mouse models of ADPKD (see PMID: 26528438 and PMID: 25139996). However, at the reviewers suggestion we have revised the title to say it increases cyst number, which is the main observation of this experiment.

Furthermore, since the pro-cystogenic effect is not conclusive and Aurka KO mice don't have pronounced cystic disease, it is an overstatement to say that Alisertib acts through Aurka vs. AurkB.

We respectfully disagree. If Alisertib acted off target on something other than AURKA to promote cysts, then it would be expected that *Inpp5e^{Δ/Δ};Aurka^{Δ/Δ}* mice would show increased cystogenesis even from their low starting baseline. However, as no change was observed (but there was a change in *Inpp5e^{Δ/Δ}*, animals, which have competent *Aurka* expression) the conclusion is that the principal pro-cystogenic effect of Alisertib is via *Aurka*. This is again consistent with previously published work studying the impact of AURKA inhibitors on ADPKD progression.

(7) The manuscript would benefit from a discussion paragraph outlining the limitations of the study, explanations for discrepancies, and future directions. For example, (a) to emphasize better that AKT is only "in part" mediating the effect as shown by striking differences in KO vs treatment and highlight how the *Aurka* and AKT data cannot be directly compared (germline loss vs treatment at existing cysts). (b) PKD1 vs *INpp5e*

differences in phenotype due to cre recombination efficiency. (c) Discussion on why AURKA is upregulated in PKD and (d) Future direction of testing other AKT inhibitors in models or testing AKT inhibitor in adult model.

We have added the following statement to the Discussion:

Given the pleiotropic role that AURKA plays in many aspects of cellular homeostasis the protein may also regulate cyst development via pathways other than AKT. Better understanding the direct and indirect mechanistic effects of AURKA on these cyst promoting pathways will be critical to understanding it's central role in disease progression.

We would note that the Discussion already indicates:

- a) that *Aurka* is transcriptionally regulated by AKT signalling and creates a feed forward loop.
- b) that intermediary AURKA binding proteins may regulate AKT activity.
- c) that AURKA may be acting a various locales within the cell (i.e not just the cilia).

(8) I don't think it is appropriate that the exact animal number per bar graph are buried in a supplemental excel table. Use single data points in the graphs or writing the "n" under/above bar graph is more appropriate to better judge the data. Also, no numbers are provided for S2h-r in the supplemental excel table. Further, no gender info per n is given.

We have now revised all graphs to include individual datapoints and have corrected any n & p-value reporting errors (e.g. lack of these measures in S2h-r). In some cases, we have included additional animals (no more than 2, as indicated) and have re-analysed the significance of findings as a result. Our re-analysis did not alter any significant findings (in fact they principally increased the significant differences reported originally). Regarding gender, we outlined in the Methods (Page 23) that "Animals of both genders were used in cohorts as most models did not show sexual dimorphism, except for Pkd1 RC mice. In the case of Pkd1 RC studies, genders were approximately 50:50 balanced across cohorts plus/minus 1 animal". We have also expanded the justification for pooling of genders with RC mice with the addition of S4a and additional details on Page 9-10 (Results) and the associated methods.

(9) S7 (now S8) to be quantified, please

This was quantified already in main Fig 6l

(10) Figure S8i (now S9i) is not referenced.

This is incorrect. Figure S9i is referenced on page 17 line 4.

REVIEWERS' COMMENTS

Reviewer #4 (Remarks to the Author):

I thank the reviewers for taking the time and detail oriented approach to elaborate on and address all my concerns. I have no further concerns. I also apologize if my comments seemed nit-picky or extensive. I truly believe this is a very strong and important study for the PKD field, hence, I wanted to assure it is in its best most robust form at time of publication. Thank you for taking the diligence to work with me towards this front. I think the manuscript has substantially improved throughout the review process.